RESEARCH COMMUNICATION

# Identification of functionally distinct fibro-inflammatory and adipogenic stromal subpopulations in visceral adipose tissue of adult mice

Chelsea Hepler[1†], Bo Shan[1], Qianbin Zhang[1], Gervaise H Henry[2], Mengle Shao[1], Lavanya Vishvanath[1], Alexandra L Ghaben[1], Angela B Mobley[3], Douglas Strand[2], Gary C Hon[4], Rana K Gupta[1]*

[1]Touchstone Diabetes Center, Department of Internal Medicine, University of Texas Southwestern Medical Center, Dallas, United States; [2]Department of Urology, University of Texas Southwestern Medical Center, Dallas, United States; [3]Department of Immunology, University of Texas Southwestern Medical Center, Dallas, United States; [4]Cecil H. and Ida Green Center for Reproductive Biology Sciences, Division of Basic Reproductive Biology Research, Department of Obstetrics and Gynecology, University of Texas Southwestern Medical Center, Dallas, United States

*For correspondence:
Rana.Gupta@UTSouthwestern.edu

†These authors contributed equally to this work

**Abstract** White adipose tissue (WAT) remodeling is dictated by coordinated interactions between adipocytes and resident stromal-vascular cells; however, the functional heterogeneity of adipose stromal cells has remained unresolved. We combined single-cell RNA-sequencing and FACS to identify and isolate functionally distinct subpopulations of PDGFRβ+ stromal cells within visceral WAT of adult mice. LY6C- CD9- PDGFRβ+ cells represent highly adipogenic visceral adipocyte precursor cells ('APCs'), whereas LY6C+ PDGFRβ+ cells represent fibro-inflammatory progenitors ('FIPs'). FIPs lack adipogenic capacity, display pro-fibrogenic/pro-inflammatory phenotypes, and can exert an anti-adipogenic effect on APCs. The pro-inflammatory phenotype of PDGFRβ+ cells is regulated, at least in part, by NR4A nuclear receptors. These data highlight the functional heterogeneity of visceral WAT perivascular cells, and provide insight into potential cell-cell interactions impacting adipogenesis and inflammation. These improved strategies to isolate FIPs and APCs from visceral WAT will facilitate the study of physiological WAT remodeling and mechanisms leading to metabolic dysfunction.

**Editorial note:** This article has been through an editorial process in which the authors decide how to respond to the issues raised during peer review. The Reviewing Editor's assessment is that all the issues have been addressed (see decision letter).

DOI: https://doi.org/10.7554/eLife.39636.001

## Introduction

White adipose tissue (WAT) represents the principle site for safe and efficient energy storage in mammals. WAT, as a whole, is considerably heterogeneous. WAT is composed of energy-storing adipocytes, various immune cell populations, vascular cells, adipocyte precursor cells (APCs), and largely uncharacterized stromal populations. The development and function of adipose tissue is highly dependent on critical interactions between adipocytes, APCs, immune cells, and endothelial cells (*Han et al., 2011*; *Hong et al., 2015*).

**eLife digest** Fat tissue, also known as white adipose tissue, specializes in storing excess calories. Much of this storage happens under the skin, but fat tissue can also build up inside the abdomen and surround organs, where it is known as 'visceral' fat. When visceral fat tissue is unhealthy, it may help diseases such as diabetes and heart disease to develop.

Unhealthy fat tissue contains enlarged fat cells, which may die from overwork. The stress this places on the surrounding tissue activates the immune system, causing inflammation and the build-up of collagen fibers around the cells (a condition known as fibrosis). Not all people develop this type of unhealthy fat tissue, but we do not yet understand why.

In many tissues, blood vessels serve as a home for several types of adult stem cells that help to rejuvenate the tissue following damage. To identify these cells, Hepler et al. analyzed the genes used by more than 3,000 cells living around the blood vessels in the visceral fat of adult mice. Recent work had already revealed that stem cells called adipocyte precursor cells live in this region. Hepler et al. now reveal the presence of a second group of cells, termed fibro-inflammatory progenitor cells (or FIPs for short).

To investigate the roles of each cell type in more detail, Hepler et al. developed a new technique to isolate the adipocyte precursor cells from other cell types. When grown in the right conditions in petri dishes, the adipocyte precursor cells were able to form new fat cells. They could also make new fat cells when transplanted into mice that lacked fat tissue. By contrast, the FIPs can suppress the activity of adipocyte precursor cells and activate immune cells. They may also help fibrosis to develop.

It is not yet clear whether FIPs are present in human fat tissue. But, if they are, understanding them in greater detail may suggest new ways to treat diabetes and heart disease in obese people.
DOI: https://doi.org/10.7554/eLife.39636.002

WAT has a unique and remarkable capacity to expand and contract in size in response to changes in demand for energy storage. In the context of positive energy balance (nutrient excess), WAT expands to meet the increased demand for energy storage, leading ultimately to the condition of obesity. The manner by which WAT expands is a critical determinant of metabolic health in obesity. It has long been appreciated that individuals who preferentially accumulate WAT in subcutaneous regions are at a relatively lower risk for developing insulin resistance when compared to equally obese individuals with central (visceral) adiposity (*Kissebah et al., 1982*; *Krotkiewski et al., 1983*). It is now widely believed that visceral and subcutaneous WAT depots represent fundamentally distinct types of WAT (*Karastergiou et al., 2013*; *Lee et al., 2013*; *Macotela et al., 2012*; *Yamamoto et al., 2010*). Indeed, visceral and subcutaneous WAT depots emanate from distinct developmental lineages (*Chau et al., 2014*).

Importantly, another clear determinant of metabolic health in obesity is manner in which individual WAT depots expand and 'remodel' (*Hepler and Gupta, 2017*; *Lee et al., 2010*). WAT 'remodeling' associated with obesity can be described as both quantitative and qualitative changes in adipocyte numbers and stromal-vascular cell composition. Pathological WAT expansion is characterized by the presence of enlarged adipocytes, excessive macrophage accumulation, and fibrosis (*Divoux et al., 2010*; *Gustafson et al., 2009*; *Hardy et al., 2011*; *Klöting and Blüher, 2014*; *Sun et al., 2013*). The prevailing hypothesis is that as 'overworked' fat cells reach their storage capacity, adipocyte death, inflammation, and fibrosis ensue (*Hepler and Gupta, 2017*; *Sun et al., 2011*). This is often associated with the deleterious accumulation of lipids in the liver, skeletal muscle, pancreas, and heart (termed 'lipotoxicity') (*Unger and Scherer, 2010*). Healthy WAT expansion occurs when adipose tissue expands through adipocyte hyperplasia (increase in adipocyte number through de novo differentiation) (*Denis and Obin, 2013*; *Kim et al., 2014*; *Klöting et al., 2010*). This is associated with a lower degree of chronic tissue inflammation and fibrosis. These adipose phenotypes of the 'metabolically healthy' obese tightly correlate with sustained insulin sensitivity in these patients. To date, the factors dictating a healthy vs. unhealthy WAT expansion in obesity remain poorly defined. In particular, the array of cell types within the adipose stromal-vascular compartment contributing to the remodeling of WAT in obesity has remained largely undefined.

The growing appreciation for the casual link between adipose tissue distribution and remodeling with systemic metabolic health has sparked considerable interest in defining the adipocyte precursors giving rise to fat cells in adults and the mechanisms controlling their differentiation in vivo (*Hepler et al., 2017*). In male C57BL/6 mice, adipose tissues expand in diet-induced obesity in a depot-selective manner. The epididymal WAT depot expands through both adipocyte hypertrophy and adipocyte hyperplasia (*Jeffery et al., 2015*; *Kim et al., 2014*; *Wang et al., 2013b*). The inguinal subcutaneous WAT depot expands almost exclusively by adipocyte hypertrophy. We recently reported that visceral adipocytes emerging in association with HFD feeding originate, at least in part, from perivascular precursors expressing *Pdgfrb* (*Vishvanath et al., 2016*). *Pdgfrb* encodes the platelet-derived growth factor receptor β chain (PDGFRβ protein) and is a widely used marker of perivascular cells (*Armulik et al., 2011*). We previously employed a pulse-chase lineage tracing mouse model to track the fate of *Pdgfrb*-expressing cells in adipose tissue. Following HFD feeding, *Pdgfrb*-expressing cells give rise to white adipocytes within visceral WAT depots (*Vishvanath et al., 2016*). The ability of these precursors to undergo de novo adipogenesis in the setting of diet-induced obesity is critical for healthy visceral WAT expansion (*Shao et al., 2018*). Inducible genetic disruption of *Pparg*, the master regulatory gene of adipocyte differentiation, in *Pdgfrb*-expressing cells leads to a loss of de novo adipogenesis from *Pdgfrb*-expressing cells in the visceral WAT depot of diet-induced obese mice; this exacerbates the pathologic remodeling of this depot (i.e. increased inflammation and fibrosis). Driving de novo adipogenesis from *Pdgfrb*-expressing cells through transgenic *Pparg* expression leads to a healthy expansion of visceral WAT (lower inflammation and small adipocytes) (*Shao et al., 2018*). The highly adipogenic subpopulation of PDGFRβ+ cells in gonadal WAT (gWAT) is quantitatively enriched in the expression of *Pparg*, as well as its upstream regulatory factor, *Zfp423* (*Gupta et al., 2012*; *Tang et al., 2008*; *Vishvanath et al., 2016*). PDGFRβ + cells enriched in these adipogenic factors express several mural cell (pericyte/smooth muscle) markers and reside directly adjacent to the endothelium in WAT blood vessels (*Gupta et al., 2012*; *Tang et al., 2008*; *Vishvanath et al., 2016*). Using *Zfp423* reporter mice (*Zfp423*[GFP] BAC transgenic mice), we revealed that PDGFRβ+ cells expressing high levels of *Zfp423* (GFP+ or *Zfp423*[High]) represent highly committed preadipocytes while *Zfp423*[Low] cells (GFP-) lacked significant adipogenic capacity, and exhibited significantly different global patterns of gene expression (*Vishvanath et al., 2016*). These observations suggested that the pool of PDGFRβ+ cells in visceral WAT is functionally heterogeneous, with cells possessing distinct cellular phenotypes.

In this study, we set out to explore the functional heterogeneity within *Pdgfrb*-expressing cells of visceral WAT from adult mice. Furthermore, our objective was to identify improved strategies to purify adipocyte precursor populations from these depots. Through single-cell RNA-sequencing, we identified functionally distinct subpopulations of *Pdgfrb*-expressing progenitor cells. We identified a unique population of cells that display fibrogenic and functional pro-inflammatory phenotypes, and lack inherent adipogenic capacity. These fibro-inflammatory progenitors (termed here as 'FIPs') can be purified by the use of commercially available antibodies (LY6C + PDGFRβ+). On the other hand, LY6C- CD9- PDGFRβ+ cells represent a distinct pool of highly adipogenic visceral adipocyte precursor cells ('APCs') that robustly differentiate spontaneously in vitro in growth media containing insulin. The frequency of these PDGFRβ+ subpopulations is highly regulated under physiological conditions. These data reveal the functional heterogeneity of perivascular progenitors within visceral WAT and provide insight into how the adipose stroma can control WAT remodeling. Moreover, the molecular profiles obtained for FIPs and APCs from visceral WAT, along with the strategies to isolate these cells, will facilitate the study of physiological WAT remodeling in vivo.

## Results

### Single-cell RNA sequencing reveals molecularly distinct *Pdgfrb*-expressing subpopulations in visceral adipose tissue

We previously derived a doxycycline-inducible (Tet-On) lineage-tracing model that allows for the indelible labeling of *Pdgfrb*-expressing perivascular cells in adipose tissue of adult mice (*Pdgfrb*[rtTA]; *TRE-Cre; Rosa26R*[mT/mG]; herein, 'MuralChaser mice') (*Vishvanath et al., 2016*). Prior to exposing animals to doxycycline, all cells within the stromal-vascular fraction (SVF) of adult gonadal WAT (gWAT) express membrane tdTomato from the *Rosa26* locus. Following 9 days of exposure to

doxycycline-containing chow diet, Cre-mediated excision of the *loxP*-flanked *tdTomato* cassette occurs in *Pdgfrb*-expressing cells, and membrane-bound GFP (mGFP) expression is constitutively activated (*Figure 1A*). As previously reported and confirmed here, FACS analysis indicated that nearly all mGFP+ cells are PDGFRβ+ as expected, and are devoid of CD45 (hematopoietic), CD31 (endothelial), and CD11b (monocyte/macrophage) expression (*Figure 1—figure supplement 1A*) (*Vishvanath et al., 2016*). Moreover, mGFP expression following transient doxycycline exposure is confined predominately to peri-endothelial cells in adult gonadal WAT (*Figure 1—figure supplement 1B*) (*Vishvanath et al., 2016*).

We set out to test the hypothesis that *Pdgfrb*-expressing perivascular cells in gonadal visceral WAT of adult mice are heterogeneous, with subpopulations harboring functionally distinct phenotypes. To this end, we performed single cell RNA-sequencing (scRNA-seq) of mGFP+ cells isolated from gWAT of lean (chow fed) 8 week-old male MuralChaser mice following 9 days of doxycycline exposure. *t*SNE analysis of 1045 cell transcriptomes revealed distinct cell clusters exhibiting unique transcriptional profiles (*Figure 1B,C*). Many of the top 20 most enriched transcripts in Cluster 1A and Cluster 1B correspond to notable genes related to adipogenesis and/or adipocyte gene expression (*Figure 1D*). In particular, the majority of cells in Clusters 1A and 1B express high levels of *Pparg*, *Fabp4*, *Hsd11b1*, and *Lpl*, indicating these clusters may represent the PDGFRβ+ APC population within visceral WAT (*Figure 1E*). Interestingly, Cluster 1B further enriches in the expression of *Pparg*, *Cebpa*, and other markers of terminal adipocyte differentiation, including *Plin1*, *Fabp5*, *Car3*, and *Cd36* (*Figure 1F*). Notably, the expression of *Adipoq*, *Retn*, and *Adrb3*, genes typically characteristic of mature adipocytes, were detected within some cells within Cluster 1B (*Figure 1F*). Unbiased gene set enrichment analysis (GSEA) revealed that cells of Cluster 1A/B enriched for gene sets related to 'adipogenesis' and cells of Cluster 1B enriched for gene signatures of 'oxidative phosphorylation,' 'adipogenesis,' and fatty acid metabolism (*Tables 1* and *2*). These data suggest Cluster 1A and 1B represent 'adipocyte precursor cells' (APCs), with Cluster 1B representing a subpopulation of APCs that are 'committed preadipocytes'.

The cells in Cluster 2 were highly enriched in the expression of genes associated with fibrosis and inflammation, including *Fn1*, *Loxl2*, *Tgfb2*, and *Ccl2* (*Figure 1D and G*). GSEA revealed the enrichment of numerous gene signatures characteristic of a fibrogenic and inflammatory phenotype, including gene sets corresponding to 'inflammatory response,' 'TGFβ signaling,' 'TNFα signaling,' and 'hypoxia' (*Table 3*). This fibro-inflammatory molecular signature of *Pdgfrb*-expressing cells suggested this subpopulation represents 'fibro-inflammatory progenitors' (herein, termed 'FIPs').

Cluster 3 was molecularly quite distinct from Clusters 1A/B and 2. Interestingly, Cluster 3 had a mesothelial-like cell (herein, 'MLCs') expression profile. Mesothelial cells are epithelial cells of mesodermal origin that form a monolayer (mesothelium) lining the visceral serosa. Mesothelial cells and mural cells share a common developmental lineage. Multiple genetic lineage tracing studies in mice indicate that various stromal cell populations within visceral tissues, including APCs, descend from embryonic mesothelial cells (*Chau et al., 2014*; *Rinkevich et al., 2012*). Mesothelial cells have been linked to multiple aspects of adipose tissue development and remodeling, including adipogenesis and inflammation (*Darimont et al., 2008*; *Gupta and Gupta, 2015*; *Mutsaers et al., 2015*). Cluster 3 was enriched for genes representing common mesothelial/epithelial markers, such as *Msln*, *Upk3b*, *Krt8*, and *Krt14* (*Figure 1D and H*). The presence of this cluster suggested that the *Pdgfrb*<sup>rtTA</sup> transgene targets at least a subset of visceral WAT associated mesothelial cells. Indeed, following transient doxycycline treatment of MuralChaser mice, a few mGFP+ cells can be observed within in the outermost epithelial layer of gonadal WAT (*Figure 1—figure supplement 1C*). Moreover, mGFP+ cells can be occasionally observed within cultures of isolated mesothelial cells obtained from gWAT of doxycycline-treated MuralChaser mice (*Figure 1—figure supplement 1D*).

We performed a second scRNA-seq analysis utilizing independently isolated mGFP+ cells from gonadal WAT MuralChaser mice (*Figure 1—figure supplement 2*). From the second scRNA-seq dataset, we again identified the same major subpopulations of *Pdgfrb*-expressing cells. All together, these scRNA-seq data reveal molecularly distinct *Pdgfrb*-expressing subpopulations in visceral adipose tissue.

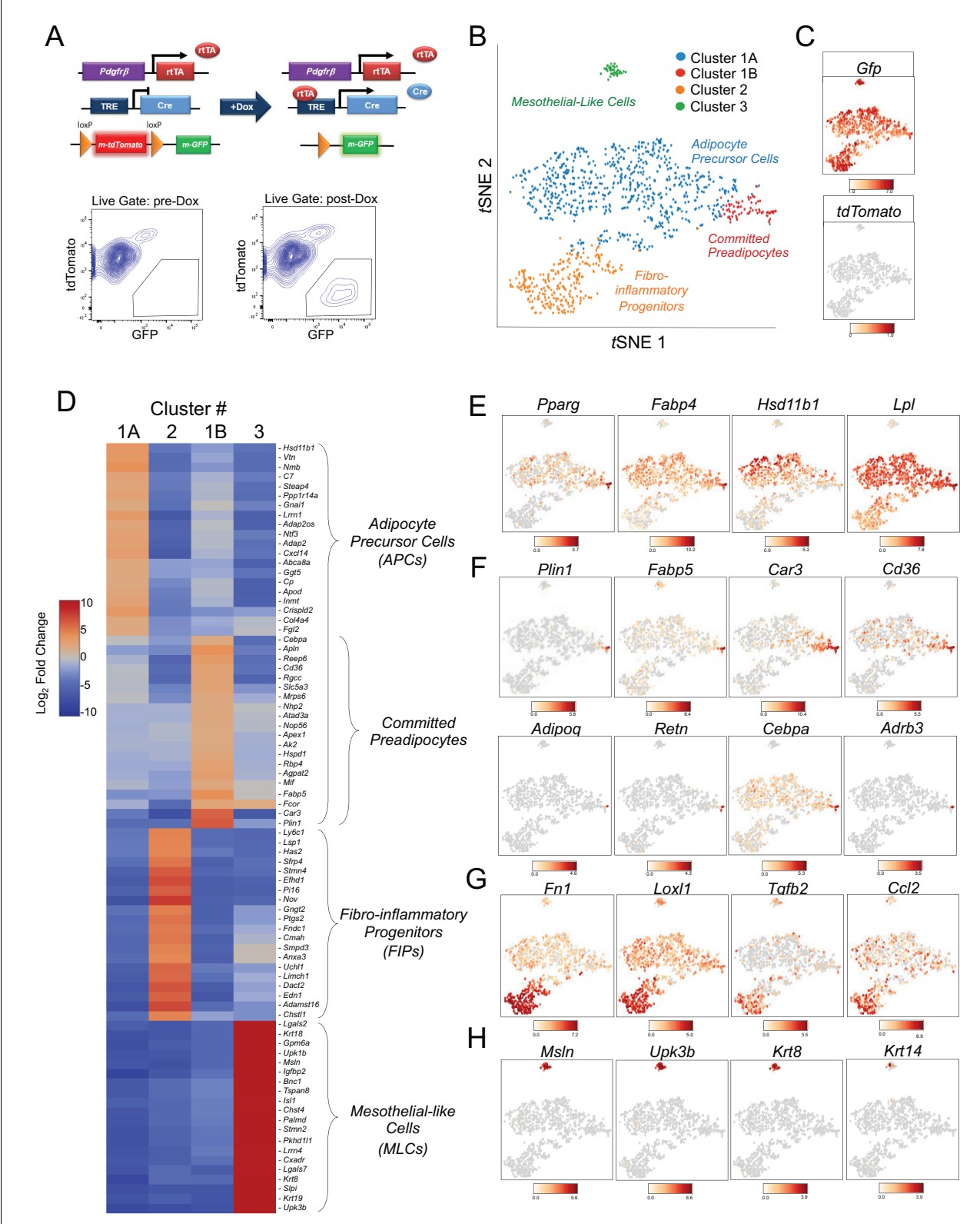

**Figure 1.** Single-cell RNA sequencing reveals molecularly distinct *Pdgfrb*-expressing subpopulations in visceral adipose tissue. (**A**) Schematic overview of the MuralChaser model: a 'Tet-On' system allowing for indelible labeling of *Pdgfrb*-expressing cells. In the absence of doxycycline (Dox), gonadal SVF cells are labeled membrane tdTomato+ and are devoid of membrane GFP expression. In the presence of Dox, rtTA activates Cre expression in *Pdgfrb*-expressing cells. Cre excises the loxP-flanked membrane tdTomato (mtdTomato) cassette and allows constitutive activation of membrane GFP

*Figure 1 continued on next page*

*Figure 1 continued*

(mGFP) reporter expression. The gating strategy shows prospective isolation of tdTomato- GFP+ cells from the stromal vascular fraction of gonadal WAT (gWAT). (B) t-distributed stochastic neighbor embedding (*tSNE*) plot of 1045 tdTomato- GFP+ cells isolated from pooled gWAT depots from five male MuralChaser mice. Equal numbers of cells were combined from five individual mice for single-cell RNA-sequencing. Clustering was generated using *k*-means = 4. See *Figure 1—source data 1*. (C) Distribution of *Gfp* and *tdTomato* expression within *tSNE* plot. Transcript counts represent Log$_2$ of gene expression. (D) Heatmap of top 20 most differentially expressed genes defining the clusters indicated in (B). See *Figure 1—source data 1*. (E) Gene expression distribution of adipocyte/adipogenesis-associated genes. (F) Gene expression distribution of genes associated with terminal adipocyte differentiation. (G) Gene expression distribution of genes associated with fibrosis and inflammation. (H) Gene expression distribution of mesothelial cell markers.

DOI: https://doi.org/10.7554/eLife.39636.003

The following source data and figure supplements are available for figure 1:

**Source data 1.** Complete list of differentially expressed genes (k-means = 4).
DOI: https://doi.org/10.7554/eLife.39636.006

**Figure supplement 1.** GFP expression in gonadal WAT of MuralChaser mice.
DOI: https://doi.org/10.7554/eLife.39636.004

**Figure supplement 2.** *tSNE* plot of 4203 tdTomato- GFP+ cells isolated from gonadal WAT of MuralChaser mice.
DOI: https://doi.org/10.7554/eLife.39636.005

## Molecularly distinct visceral WAT PDGFRβ+ subpopulations can be isolated by FACS

Next, we developed a strategy to isolate these molecularly distinct cell populations by flow cytometry from wild type mice. For this purpose, we treated Cluster 1A and 1B as one broad 'APC' population (*Figure 2A*). Candidate cell surface markers were selected on the basis of their corresponding gene expression in the three PDGFRβ+ subpopulations and the availability of commercial antibodies suitable for FACS. Of note, *Ly6c1* expression was abundant in FIPs but not APCs (*Figure 2B*). The expression of *Cd9*, a recently described marker of fibrogenic cells (*Marcelin et al., 2017*), was abundantly expressed in both the FIPs and MLCs (*Figure 2B*). Therefore, we isolated the three populations based on these markers using fluorescence-activated cell sorting. PDGFRβ+ cells (CD31- and CD45-) were subdivided on the basis of LY6C and CD9 immunoreactivity (*Figure 2B,C*). Three distinct subpopulations of PDGFRβ+ cells were apparent: LY6C- CD9- (APCs), LY6C+ (FIPs), and LY6C- CD9+ (MLCs) cells (*Figure 2C*). Flow cytometry analysis consistently revealed that LY6C+ PDGFRβ+ cells were more abundant than LY6C- CD9- PDGFRβ+ cells and Ly6C- CD9+ PDGFRβ+ cells (*Figure 2D*). Importantly, gene expression analysis by qPCR revealed that LY6C- CD9- PDGFRβ+

**Table 1.** Gene sets enriched in APCs (Cluster 1A/B).

| Gene set name | Gene set description | FDR q-value | Enriched genes |
|---|---|---|---|
| HALLMARK_XENOBIOTIC_METABOLISM | Genes encoding proteins involved in processing of drugs and other xenobiotics. | 0.008879008 | APOE, IGF1, NDRG2, VTN, HSD11B1, ENPEP, POR, TNFRSF1A, SLC1A5, JUP, PMM1, CD36, PTGES, FAH, FMO1, HMOX1, GCNT2, ABCD2, ECH1, GSTA3, AOX1, IL1R1, GABARAPL1, ID2, CASP6, CSAD, MPP2, DDT, GSTO1, ALDH2, TMEM176B, GSTT2, CYP27A1, CYB5A, SMOX, FBLN1, MCCC2, ELOVL5, NQO1, PDK4, ALAS1, ATP2A2, RBP4, TMEM97 |
| HALLMARK_ADIPOGENESIS | Genes up-regulated during adipocyte differentiation (adipogenesis). | 0.033845212 | GPX3, SPARCL1, COL15A1, APOE, LPL, COL4A1, MYLK, CMBL, LIFR, SDPR, EPHX2, PPARG, POR, MRAP, REEP6, SLC1A5, ENPP2, ANGPTL4, CD302, FABP4, ANGPT1, GPHN, CD36, SLC27A1, RAB34, LIPE, PTGER3, IFNGR1, FAH, ALDOA, SULT1A1, FZD4, SCP2, TST, ECH1, SLC19A1, ADCY6, TANK, CS, ACADM, DDT, UBC, MCCC1, ALDH2, BCKDHA, AGPAT3, DBT, JAGN1, MGST3, ADIPOR2, SLC5A6, DNAJC15, GPAM, PIM3, CYP4B1, RETSAT, ITGA7, SLC25A10, SCARB1 |
| HALLMARK_IL6_JAK_STAT3_SIGNALING | Genes up-regulated by IL6 via STAT3, e.g., during acute phase response. | 0.08689988 | SOCS3, JUN, CNTFR, TNFRSF1A, CD38, PIM1, OSMR, CD36, IFNGR1, SOCS1, IL17RA, MYD88, HMOX1, IRF1, STAT3, IL1R1, STAT2 |

DOI: https://doi.org/10.7554/eLife.39636.007

**Table 2.** Gene sets enriched in committed preadipocytes (Cluster 1B).

| Gene set name | Gene set description | FDR q-value | Enriched genes |
|---|---|---|---|
| HALLMARK_MYC_TARGETS_V2 | A subgroup of genes regulated by MYC - version 2. | 0 | SRM, GNL3, NOLC1, HSPE1, NIP7, HSPD1, PA2G4, NPM1, CDK4, PPAN, MYBBP1A, RCL1, PUS1, PHB, WDR43, HK2, WDR74, SLC19A1, GRWD1, EXOSC5, PES1, PRMT3, DDX18, TMEM97, IMP4, UNG, UTP20, LAS1L, MPHOSPH10, PPRC1, NOC4L, TBRG4, BYSL, IPO4, TFB2M |
| HALLMARK_OXIDATIVE_PHOSPHORYLATION | Genes encoding proteins involved in oxidative phosphorylation. | 0 | ATP5G1, NNT, COX8A, TIMM13, TIMM10, LDHA, CYCS, TOMM70A, UQCRQ, COX7C, CYC1, COX7A2, ATP5G2, TIMM50, ATP5E, NDUFA4, NDUFAB1, SLC25A5, ATP5L, SLC25A4, PHB2, ACAT1, ATP5J, ATP5C1, CS, NDUFB8, NDUFB2, GRPEL1, UQCRFS1, IDH3A, NDUFV2, COX5A, NDUFC2, MRPS15, NDUFB4, POR, ECHS1, ATP5B, MRPS12, COX7B, LDHB, COX4I1, ATP5D, MRPL15, COX6B1, UQCRH, MDH2, SLC25A3, TIMM9, ATP5G3, NDUFB5, PRDX3, NDUFA2, ATP5A1, MRPS30, ATP5H, NDUFA7, NDUFC1, COX5B, PDHB, ATP5F1, MAOB, BAX, NDUFA3, GPX4, NDUFS8, VDAC2, COX6C, POLR2F, NDUFS3, COX6A1, NDUFS2, UQCRB, TIMM17A, ACADM, NDUFS7, ATP5O, MRPL11, IDH1, MRPL35, SUCLG1, HCCS, SDHD, MRPL34, MRPS11, NDUFB7, VDAC1, ATP5J2, NDUFA8, GOT2, OXA1L, SLC25A11, NDUFS6, NDUFA6, ETFB, IMMT, HTRA2, MTRR, FXN, SDHB, ACO2, FDX1, NDUFB6, DLAT, PMPCA, DLD, IDH2, AFG3L2, ETFDH, MTX2, TIMM8B, RETSAT, COX7A2L, TOMM22, NDUFA5, SUCLA2, UQCRC1, ALDH6A1, RHOT1, ECH1, SURF1, ATP6V1G1, VDAC3, PDHX, LRPPRC, UQCRC2, HADHB |
| HALLMARK_ADIPOGENESIS | Genes up-regulated during adipocyte differentiation (adipogenesis). | 0 | REEP6, COL15A1, MYLK, APOE, COX8A, PIM3, CMBL, UQCRQ, LPL, SLC1A5, CYC1, PPARG, NDUFAB1, TKT, YWHAG, CS, DBT, GRPEL1, IDH3A, SCP2, SLC25A10, POR, ECHS1, FZD4, G3BP2, COX7B, SLC19A1, AK2, MRPL15, JAGN1, ENPP2, MDH2, ALDOA, PRDX3, MRAP, RAB34, DDT, MTCH2, HADH, PTGER3, LIPE, CPT2, REEP5, MCCC1, ANGPT1, GPX4, AGPAT3, NDUFS3, COX6A1, TANK, ACADM, SCARB1, ATP5O, ADCY6, GPX3, IDH1, SUCLG1, PEX14, SPARCL1, SDPR, PREB, GHITM, ALDH2, ADIPOR2, NDUFB7, EPHX2, ACADS, DNAJC15, GPHN, HIBCH, FAM73B, CHUK, VEGFB, ETFB, IMMT, ACOX1, RREB1, QDPR, FABP4, ACLY, ELOVL6, SDHB, PFKL, ACO2, RETN, CAT, PTCD3, DLAT, DLD, TST, CD36, DHRS7B, ITSN1, RETSAT, NDUFA5, UQCRC1, UBQLN1, DNAJB9, ECH1, SLC27A1 |
| HALLMARK_MYC_TARGETS_V1 | A subgroup of genes regulated by MYC - version 1 (v1). | 0 | RPLP0, SRM, RPL6, GNL3, RPS2, RPL18, CNBP, RPS5, APEX1, RPL14, RPS6, RANBP1, SERBP1, ERH, C1QBP, RPL34, NOLC1, HSPE1, HSPD1, PABPC1, SET, LDHA, EIF4A1, RPS3, PA2G4, SNRPD1, RSL1D1, TOMM70A, RAN, DDX21, NPM1, EIF2S2, CYC1, PABPC4, CDK4, IMPDH2, FBL, NAP1L1, NDUFAB1, RPL22, ABCE1, PHB2, HDGF, SNRPD2, LSM7, RPS10, HSP90AB1, PHB, CCT2, PPM1G, SNRPD3, SYNCRIP, PCBP1, CCT3, LSM2, EPRS, NME1, EIF2S1, GSPT1, COX5A, CCT7, CCT5, TUFM, U2AF1, PPIA, TCP1, ODC1, POLE3, ACP1, EEF1B2, TARDBP, YWHAE, SLC25A3, EIF1AX, SNRPA1, ETF1, SRPK1, PSMD7, PRDX3, SMARCC1, RAD23B, CCT4, RNPS1, FAM120A, RUVBL2, TXNL4A, EIF4E, KARS, PTGES3, GLO1, DDX18, MCM7, CANX, DUT, PRPF31, UBE2L3, KPNB1, NCBP1, SNRPA, POLD2, PSMA7, EIF4G2, PSMB2, PRPS2, DHX15, SSBP1, CLNS1A, PSMB3, PGK1, XPOT, STARD7, H2AFZ, ILF2, VDAC1, SSB, CTPS, GOT2, MRPS18B, SNRPG, COPS5, MRPL9, PSMA2, CAD, PSMA4, TRIM28, IARS, SF3B3, PSMD14, SNRPB2, UBE2E1, NCBP2, PWP1, YWHAQ, PSMD8, AP3S1, RFC4, HDDC2, PSMA6, XPO1, VDAC3, PSMC4, CDK2, USP1, MYC, PCNA, MRPL23 |
| HALLMARK_DNA_REPAIR | Genes involved in DNA repair. | 0.001391793 | AK1, TMED2, BOLA2, IMPDH2, POLR1D, SAC3D1, APRT, NUDT9, NME1, NUDT21, SSRP1, RAE1, ADRM1, GTF2A2, GUK1, POLR2D, GTF2H5, GPX4, POLR2F, MPG, DUT, SEC61A1, ADCY6, POLR2E, POLE4, RBX1, NT5C3, POLR1C, AK3, POLR2C, TAF10, GTF2H1, RNMT, DDB1, NME4, NFX1, POLR3GL, EIF1B, POLR2G, NCBP2, POLR2K, POLR2H, SURF1, ERCC8, TSG101, RFC4, RFC5, PCNA, UPF3B, POLR2I, RAD51, ITPA, EDF1, PRIM1, DAD1, TAF12, GTF2F1, POLD3, TCEB3, DCTN4, ARL6IP1, POLA1 |
| HALLMARK_MTORC1_SIGNALING | Genes up-regulated through activation of mTORC1 complex. | 0.001159828 | PSAT1, ATP5G1, HSPE1, HSPD1, LDHA, TOMM40, SLC1A5, EIF2S2, ENO1, EEF1E1, PHGDH, ARPC5L, SQLE, EPRS, HSPA4, PPIA, PSME3, HK2, GAPDH, MTHFD2, ETF1, ALDOA, PDAP1, PPA1, XBP1, ABCF2, BCAT1, UBE2D3, CACYBP, CYB5B, PSMA3, SLC7A5, TXNRD1, CANX, INSIG1, TMEM97, IDH1, HMBS, SSR1, PSMB5, ADIPOR2, PGK1, SERPINH1, UNG, PLOD2, PSPH, PRDX1, POLR3G, RPN1, DAPP1, IMMT, SLC2A1, QDPR, ACLY, ELOVL6, ATP2A2, PFKL, GTF2H1, COPS5, LDLR, SHMT2, UFM1, PSMA4, FDXR, TCEA1, GMPS, IDI1, PSMD12, ELOVL5, PSMD14, MAP2K3, PITPNB, MLLT11, TPI1, GSK3B, M6PR, PSMC4, ME1, NUP205, SLC2A3, NUFIP1, GSR, UCHL5, HMGCR |
| HALLMARK_FATTY_ACID_METABOLISM | Genes encoding proteins involved in metabolism of fatty acids. | 0.00329518 | REEP6, MIF, APEX1, LDHA, AOC3, FASN, SUCLG2, ECHS1, ODC1, MDH2, ALDOA, HADH, PDHB, BCKDHB, CPT2, ACADM, SETD8, ADSL, IDH1, SUCLG1, HCCS, SDHD, ADIPOR2, ERP29, H2AFZ, ACADS, HIBCH, PRDX6, ACOX1, GSTZ1, ACO2, GRHPR, G0S2, DLD, CD36, ACSL1, IDI1, ELOVL5, ETFDH, CCDC58, RETSAT, METAP1, SUCLA2, ECH1, HSP90AA1, HSPH1, MCEE, HADHB, ME1, GCDH, IDH3B, CRAT, SDHC, MLYCD, AQP7, DLST, HSD17B7, HMGCS1, SMS, GPD1, RDH11, ACADVL, NSDHL, HMGCL, DECR1, ACSL5, UROS |

*Table 2 continued on next page*

*Table 2 continued*

| Gene set name | Gene set description | FDR q-value | Enriched genes |
|---|---|---|---|
| HALLMARK_ PEROXISOME | Genes annotated by the GO term GO:0005777. A small, membrane-bounded organelle that uses dioxygen (O2) to oxidize organic molecules; contains some enzymes that produce and others that degrade hydrogen peroxide (H2O2). | 0.002883282 | CNBP, PABPC1, SLC25A4, SCP2, SMARCC1, PEX11A, FDPS, SLC35B2, SOD2, IDH1, PEX14, EPHX2, CTPS, GNPAT, PRDX1, PEX13, NUDT19, ACOX1, CTBP1, CAT, IDH2, ACSL1, IDI1, ELOVL5, RETSAT, ECH1, ABCD3, SLC25A17, PEX5, CDK7, CRAT, MLYCD, PEX11B, HRAS, DHRS3, ISOC1, RDH11, ABCD2, HMGCL, ACSL5, SLC23A2, SOD1, TOP2A, CRABP1 |
| HALLMARK_ E2F_TARGETS | Genes encoding cell cycle related targets of E2F transcription factors. | 0.004273379 | RANBP1, NOLC1, CKS1B, PA2G4, RAN, LYAR, CDK4, NAP1L1, SYNCRIP, NME1, EIF2S1, GSPT1, PHF5A, MTHFD2, AK2, NUDT21, SSRP1, SNRPB, TIPIN, UBE2S, IPO7, PNN, MCM7, SHMT1, DUT, H2AFX, NUP153, HN1, POLD2, POLE4, HMGB3, H2AFZ, UNG, CTPS, HELLS, PAICS, CENPM, ILF3, RBBP7, PSIP1, RAD1, TBRG4, NASP, PRPS1, PSMC3IP, TK1, BRMS1L, RAD51AP1, CDKN2A, CTCF, RAD50, POP7, XPO1, TCF19, ASF1A, CDKN2C, USP1, NUP205, MYC, PCNA, POLE, PPP1R8, ASF1B, SMC1A, ATAD2, DIAPH3, MCM5, CCNB2, DEK, RFC1, XRCC6, BRCA2, CSE1L, EZH2, ANP32E, POLD3, MCM2, SMC6, MCM6, RQCD1, DONSON, ZW10, CKS2, BRCA1, MRE11A, RPA3, KIF22, PLK4, BIRC5, CDC25A, GINS1, CDCA3, KPNA2, HMMR, SMC4, CCNE1, MXD3, EXOSC8, RFC2, MLH1, TRIP13, TOP2A, MAD2L1 |
| HALLMARK_ UNFOLDED_ PROTEIN_RESPONSE | Genes up-regulated during unfolded protein response, a cellular stress response related to the endoplasmic reticulum. | 0.005032035 | PSAT1, RPS14, NOLC1, CKS1B, EIF4A1, EEF2, NPM1, DKC1, LSM4, EIF4EBP1, EIF2S1, EXOSC1, MTHFD2, SDAD1, XBP1, EXOSC5, EIF4E, EIF4G1, SLC7A5, EXOSC2, H2AFX, CEBPG, SSR1, PREB, XPOT, BANF1, DDX10, EXOSC4, FUS, PARN, TARS, LSM1, SRPRB, IARS, SPCS1, DNAJB9, BAG3, EIF4A2 |
| HALLMARK_G2M_ CHECKPOINT | Genes involved in the G2/M checkpoint, as in progression through the cell division cycle. | 0.005412505 | NCL, NOLC1, CKS1B, SNRPD1, CDK4, DKC1, DTYMK, UCK2, SYNCRIP, SQLE, GSPT1, HSPA8, ODC1, EWSR1, SMARCC1, RAD23B, HMGN2, UBE2S, PRPF4B, DR1, PRMT5, AMD1, SLC7A5, SETD8, H2AFX, HN1, KPNB1, HMGB3, SFPQ, H2AFZ, ILF3, TNPO2, SLC7A1, TOP1, NASP, CBX1, NUP50, CASP8AP2, E2F4, CTCF, XPO1, CDKN2C, MYC, CUL4A, POLE, CCNT1, YTHDC1, SMC1A, MCM5, CCNB2, BRCA2, CASC5, KATNA1, POLQ, EZH2, CUL1, MCM2, ODF2, MTF2, MCM6, WHSC1, NEK2, E2F1, SMC2, SS18, CKS2, E2F3, KIF22, PLK4, HIF1A, EXO1, BIRC5, H2AFV, CDC25A, KPNA2, CHAF1A, PAFAH1B1, HMMR, SMC4, PBK, TROAP, GINS2, CENPF, CCNA2, RBM14, TOP2A, MAD2L1, KIF11, STMN1, BUB3, DBF4, RPA2, TPX2, RBL1, BARD1, UPF1, CENPE, ATRX, KIF5B, HIRA, PRC1, CCND1, CDC27, CHEK1, CENPA, SUV39H1, MNAT1, STIL, POLA2, TFDP1, FBXO5, PURA, MKI67, AURKA, UBE2C, EGF, CDC25B, ZAK, TMPO, CUL5, MCM3, WRN, MYBL2, RAD54L, LIG3, TTK, SMAD3, RACGAP1 |
| HALLMARK_REACTIVE_ OXIGEN_SPECIES_ PATHWAY | Genes up-regulated by reactive oxigen species (ROS). | 0.011108679 | MGST1, NDUFB4, SOD2, GPX4, TXNRD1, NDUFS2, GPX3, PRDX2, PRDX6, PRDX1, NDUFA6, PPP2R4, CAT, MSRA, GLRX2 |
| HALLMARK_PI3K_ AKT_MTOR_ SIGNALING | Genes up-regulated by activation of the PI3K/AKT/mTOR pathway. | 0.012494773 | PLA2G12A, PTEN, CDK4, PRKAR2A, PPP1CA, PFN1, PIN1, UBE2D3, EIF4E, PLCB1, UBE2N, AKT1S1, AKT1, RPS6KA3, TNFRSF1A, DAPP1, SLC2A1, YWHAB, PPP2R1B, MKNK2, CFL1, ECSIT, MAPKAP1, MAP2K3, PLCG1, ATF1, GSK3B, RAF1, CDK2, MAP3K7, ARHGDIA, HRAS, CAB39L, RIPK1, E2F1, CALR, AP2M1, MYD88, CSNK2B, ARF1, PTPN11, PAK4, SMAD2 |
| HALLMARK_ XENOBIOTIC_ METABOLISM | Genes encoding proteins involved in processing of drugs and other xenobiotics. | 0.018676866 | IGF1, APOE, CSAD, SLC1A5, GSTO1, RBP4, PMM1, POR, ENPEP, ACP1, NDRG2, DDT, BCAT1, KARS, PTGES3, IL1R1, TMEM97, IDH1, PTGES, ALDH2, MCCC2, TNFRSF1A, ACOX1, MTHFD1, ATP2A2, TPST1, PGD, ACO2, CAT, SHMT2, IGFBP4, GART, CD36, ELOVL5, ETFDH, RETSAT, SSR3, ADH5, DDAH2, ECH1 |

DOI: https://doi.org/10.7554/eLife.39636.008

**Table 3.** Gene sets enriched in FIPs.

| Gene set name | Gene set description | FDR q-value | Enriched genes |
|---|---|---|---|
| HALLMARK_PANCREAS_BETA_CELLS | Genes specifically up-regulated in pancreatic beta cells. | 0 | DPP4, LMO2, SRP9, SRP14 |
| HALLMARK_INFLAMMATORY_RESPONSE | Genes annotated by the GO term GO:0006954. The immediate defensive reaction to infection or injury caused by chemical or physical agents. The process is characterized by local vasodilation, extravasation of plasma into intercellular spaces and accumulation of white blood cells and macrophages. | 5.02E-04 | AXL, CD55, HAS2, ITGB3, EMP3, IRF7, TNFRSF1B, NFKBIA, EDN1, DCBLD2, ATP2B1, CCL2, SRI, IL18, BST2, ADORA2B, CSF1, TNFAIP6, ADM, ITGA5, CCL7, TLR2, TPBG, HIF1A, PDPN, TAPBP, ABI1, KLF6, NFKB1, SERPINE1, GNAI3, RHOG, CCRL2, SLC7A1, ABCA1, SLC4A4, CDKN1A, GPC3, PVR, PLAUR, IFNGR2, IL18R1, RELA, IL6, P2RY2, EIF2AK2, TIMP1, MMP14, GCH1, LIF, CXCL10, KIF1B |
| HALLMARK_UV_RESPONSE_DN | Genes down-regulated in response to ultraviolet (UV) radiation. | 0.00149668 | TGFBR2, EFEMP1, CYR61, FYN, CDON, HAS2, LAMC1, ANXA4, ITGB3, MGLL, ANXA2, PMP22, COL1A1, APBB2, ATP2B1, VLDLR, SRI, NR3C1, FBLN5, ADORA2B, COL1A2, COL3A1, PDLIM5, FZD2, IGFBP5, DUSP1, ADD3, SMAD7, SYNE1, CITED2, TGFBR3, NOTCH2, NFKB1, SERPINE1, ATRX, SDC2, SLC7A1, IGF1R, VAV2, CDKN1B, NEK7 |
| HALLMARK_COAGULATION | Genes encoding components of blood coagulation system; also up-regulated in platelets. | 0.00112251 | FN1, FBN1, PRSS23, DPP4, S100A13, FYN, BMP1, ANXA1, ITGB3, GDA, SPARC, CD9, PLAT, RAC1, ARF4, WDR1, CAPN2, ADAM9, SERPINE1, PECAM1, MAFF, DUSP14, KLF7, GNB2, HMGCS2, GNG12, TIMP1, TIMP3, MMP14 |
| HALLMARK_TGF_BETA_SIGNALING | Genes up-regulated in response to TGFB1. | 8.98E-04 | RHOA, SPTBN1, FKBP1A, BMP2, SKIL, SMURF2, CTNNB1, SMURF1, CDKN1C, SKI, SMAD7, BMPR2, SERPINE1, TGFBR1, ID3, IFNGR2, SMAD1, ACVR1, KLF10 |
| HALLMARK_EPITHELIAL_MESENCHYMAL_TRANSITION | Genes defining epithelial-mesenchymal transition, as in wound healing, fibrosis and metastasis. | 7.48E-04 | FN1, PCOLCE2, MFAP5, FBN1, FSTL1, LOXL1, CYR61, BMP1, THY1, LAMC1, ITGB3, EMP3, ECM1, SFRP4, DPYSL3, LOXL2, TPM4, SPARC, CAPG, CALU, LGALS1, PMP22, BASP1, TNFRSF11B, COL1A1, ITGB5, POSTN, FGF2, ANPEP, FLNA, PRRX1, CXCL1, EFEMP2, THBS2, TPM1, ITGAV, PPIB, TNFRSF12A, PDLIM4, SAT1, FBLN5, COL1A2, PTHLH, DST, LAMC2, COL3A1, IGFBP4, TPM2, ITGA5, COL16A1, ITGB1, WIPF1, FBN2, CALD1, PFN2, FZD8, TGFBR3, NOTCH2, SERPINE1, COL12A1 |
| HALLMARK_APICAL_JUNCTION | Genes encoding components of apical junction complex. | 7.89E-04 | FBN1, CD34, ACTG1, ADRA1B, THBS3, BMP1, THY1, MYH10, SIRPA, ZYX, CNN2, FLNC, TNFRSF11B, ARPC2, YWHAH, EPB41L2, LIMA1, MSN, ITGA9, PFN1, ACTB, VCL, PVRL3, RSU1, LAMC2, PARVA, COL16A1, ITGB1, PVRL1, CTNNA1, ADAM9, ADAM15, GAMT, PECAM1, PVRL4, CD276, VAV2, RRAS |
| HALLMARK_ALLOGRAFT_REJECTION | Genes up-regulated during transplant rejection. | 9.30E-04 | CD47, THY1, RPL39, TGFB2, IRF7, CAPG, RPS9, FLNA, B2M, RPS19, CCL2, RPL9, CSK, GALNT1, IL18, CSF1, CCND3, INHBB, CCL7, TLR2, HIF1A, TAPBP, ELF4, IRF4, ABI1, PSMB10, CD80, IFNGR2, IL6, NPM1, UBE2D1, TIMP1 |
| HALLMARK_APICAL_SURFACE | Genes encoding proteins over-represented on the apical surface of epithelial cells, e.g., important for cell polarity (apical area). | 0.002962323 | SULF2, THY1, HSPB1, DCBLD2, EFNA5, ADAM10, PLAUR, ATP8B1 |
| HALLMARK_MITOTIC_SPINDLE | Genes important for mitotic spindle assembly. | 0.002666091 | MARCKS, FLNB, MYH10, TRIO, SPTBN1, FLNA, EPB41L2, SPTAN1, MAPRE1, RALBP1, CAPZB, ARHGAP29, ABL1, VCL, NIN, DST, ARF6, PDLIM5, CLASP1, YWHAE, KIFAP3, PXN, LMNB1, ARHGDIA, ABI1, NOTCH2, BIN1, DOCK4, KIF5B, PKD2, MYO1E, HOOK3, FARP1, WASF2, DYNC1H1, PREX1, MYH9, CKAP5, SMC3, SOS1, ITSN1, DYNLL2, CDK5RAP2, SMC1A, ARHGEF3, ESPL1, KIF1B, NEDD9, TIAM1, PPP4R2, ROCK1, PALLD, CD2AP, WASF1, CDC42BPA, RASA2, CDC42EP2, RHOT2, ALMS1, APC, PCM1, CDC27 |

*Table 3 continued on next page*

Table 3 continued

| Gene set name | Gene set description | FDR q-value | Enriched genes |
|---|---|---|---|
| HALLMARK_ COMPLEMENT | Genes encoding components of the complement system, which is part of the innate immune system. | 0.003505879 | FN1, DPP4, CD55, TIMP2, ATOX1, S100A13, GNGT2, FYN, KIF2A, IRF7, PLA2G4A, PLAT, CXCL1, CALM1, EHD1, PFN1, ADAM9, IRF2, SERPINE1, GNAI3, RHOG, PRCP, MAFF, GCA, DOCK4, PLAUR, GNB2, IL6, CEBPB, TIMP1, GNAI2, XPNPEP1, MMP14 |
| HALLMARK_PROTEIN_ SECRETION | Genes involved in protein secretion pathway. | 0.004820185 | GNAS, PAM, ATP1A1, CLTA, ADAM10, DST, AP2B1, VAMP3, SSPN, RPS6KA3, MAPK1, SCRN1, AP3S1, ARFGAP3, SOD1, ABCA1, AP2S1, COPE, SNX2, ARFIP1, AP2M1, ARCN1, COPB1, ANP32E, LMAN1, CLTC, ERGIC3, DNM1L, RAB22A, TMED10, KIF1B, BET1, RAB14, COPB2, TSG101, AP3B1, STX12, GOLGA4, VPS4B, ARF1, MON2, RER1 |
| HALLMARK_TNFA_ SIGNALING_VIA_NFKB | Genes regulated by NF-kB in response to TNF. | 0.008500786 | GFPT2, NR4A1, MARCKS, CYR61, PTGS2, SPSB1, NFKBIA, NR4A3, NFE2L2, EDN1, FOSL2, KLF2, CXCL1, ATP2B1, EIF1, PLK2, CCL2, B4GALT5, BMP2, EHD1, CCNL1, IER3, IL18, SAT1, NFIL3, CSF1, TNFAIP6, PDLIM5, NR4A2, TLR2, DUSP1, TRIP10, JAG1, RELB, PER1, IER2, TUBB2A, IER5, CXCL2, KLF6, NFKB1, SERPINE1, CCRL2, NFKBIE, MAFF, ABCA1, CDKN1A, KLF4, PLAUR, CD80, NFKB2, IFNGR2, RELA, IL6, CEBPB, GEM, FOSL1, IFIT2, DNAJB4, KLF10, ETS2, DDX58, GCH1, LIF |
| HALLMARK_HYPOXIA | Genes up-regulated in response to low oxygen levels (hypoxia). | 0.01474173 | PRDX5, CYR61, AKAP12, EXT1, CSRP2, PLAC8, UGP2, NDRG1, PTRF, ANXA2, PRKCDBP, PAM, HAS1, FOSL2, VLDLR, SLC6A6, HS3ST1, NAGK, ERRFI1, NR3C1, IER3, NFIL3, ADORA2B, ADM, CDKN1C, DUSP1, TPBG, DTNA, TPST2, CITED2, HK1, WSB1, KLF6, SERPINE1, GAPDH, SDC2, MAFF, AMPD3, PFKP, CDKN1A, CTGF, GPC3, IDS, PLAUR, KLF7, CDKN1B, PGM1, IL6, SULT2B1, TES, XPNPEP1, MYH9, HK2 |

DOI: https://doi.org/10.7554/eLife.39636.009

cells were enriched in the expression of genes that defined the APC population (Cluster 1) (*Figure 2E,F*). LY6C+ PDGFRβ+ cells enriched for the mRNAs that initially defined the FIPs (Cluster 2) (*Figure 2E,G*), and LY6C- CD9+ PDGFRβ+ cells expressed the mesothelial/epithelial markers that defined Cluster 3 (*Figure 2E,H*). Collectively, these data provide independent validation of the scRNA-seq data of genetically labeled *Pdgfrb*-expressing cells, and establish a method for isolating PDGFRβ+ subpopulations from gWAT of adult wild type mice using commercially available antibodies.

## LY6C- CD9- PDGFRβ+ cells are functional visceral white adipocyte precursors

The global molecular signature of LY6C- CD9- PDGFRβ+ cells (Cluster 1) suggests this population represents APCs. Indeed, freshly sorted LY6C- CD9- PDGFRβ+ cells are enriched in *Pparg* expression when compared to LY6C+ PDGFRβ+ cells (*Figure 3—figure supplement 1A*). We explored this hypothesis by testing the ability of these subpopulations to undergo adipocyte differentiation in vitro. We isolated and cultured all three subpopulations in growth medium containing 2% FBS and 1% ITS (insulin, transferrin, selenium). These represent culture conditions that we previously established for growth and differentiation of gWAT-derived PDGFRβ+ cells (*Vishvanath et al., 2016*). Under these growth conditions, LY6C+ PDGFRβ+ cells proliferate at a greater rate than LY6C- CD9- PDGFRβ+ cells; however, the two subpopulations appear morphologically indistinguishable, with both populations appearing fibroblast-like until reaching confluence (*Figure 3—figure supplement 1B,C,E*). LY6C- CD9+ PDGFRβ+ cells (MLCs) grow to confluence and adopt a cobblestone-like morphology characteristic of cultured mesothelial cells (*Figure 3—figure supplement 1D*). Remarkably, upon reaching confluence, only LY6C- CD9- PDGFRβ+ cells (APCs) underwent spontaneous

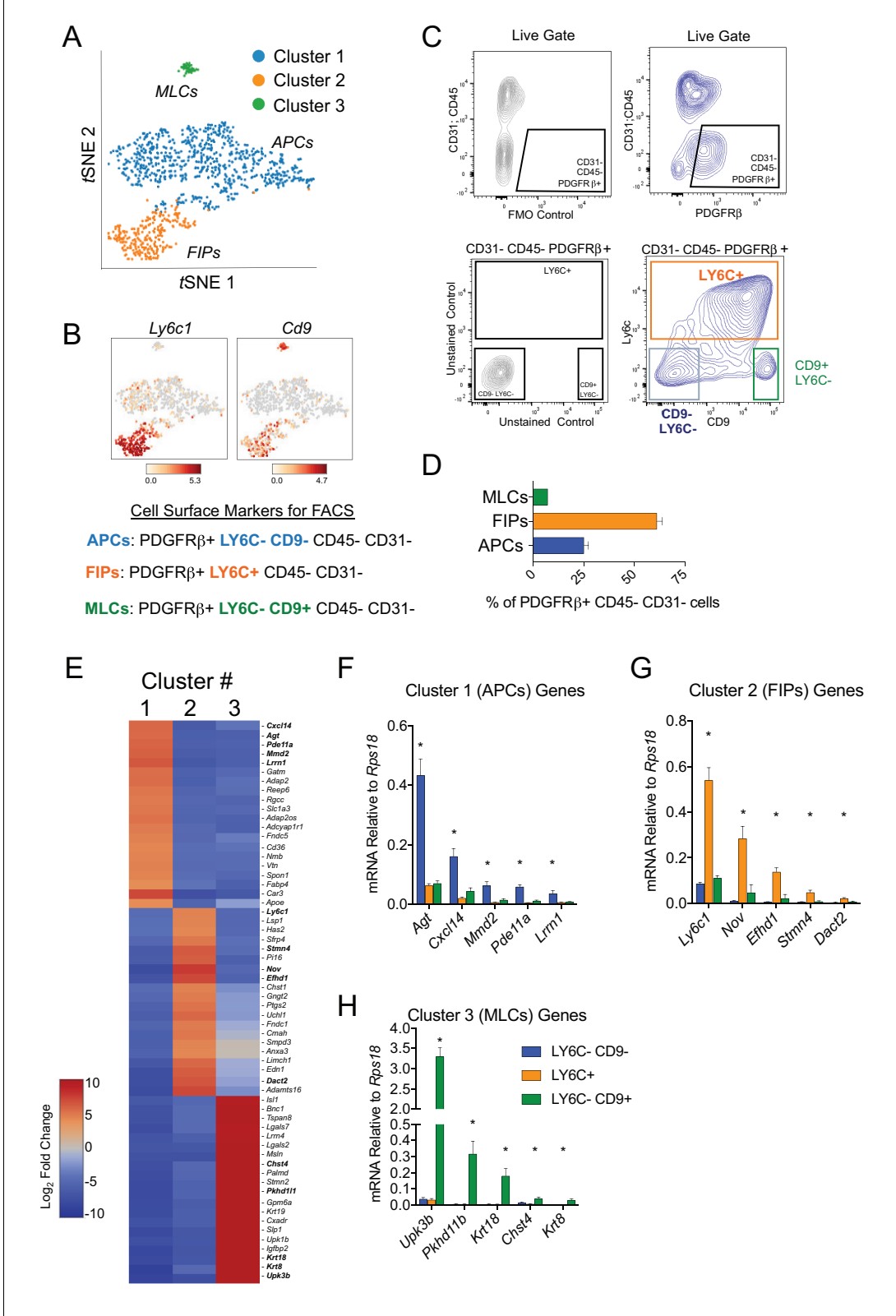

**Figure 2.** Isolation of gonadal WAT PDGFRβ+ subpopulations by FACS. (**A**) *tSNE* plot of cells from *Figure 1B* with k-means = 3 clustering. See *Figure 2—source data 1*. (**B**) Distribution of *Ly6c1* and *Cd9* expression within *tSNE* plot. Transcript counts represent Log₂ of gene expression. (**C**) Fluorescence-activated cell sorting (FACS) gating strategy to isolate indicated PDGFRβ+ CD31 CD45- subpopulations from gWAT. (**D**) Frequency of APCs, FIPs, and MLCs in gonadal WAT isolated from lean male 8 week old C57BL/6 mice. Frequencies were quantified based on the gating strategy

*Figure 2 continued on next page*

*Figure 2 continued*

shown in (C). n = 6. (E) Heatmap of top 20 most differentially expressed genes that define the clusters depicted in (A). See *Figure 2—source data 1*. (F) mRNA levels of Cluster 1 genes in freshly isolated APCs (LY6C- CD9-), FIPs (LY6C+), and MLCs (LY6C- CD9-), obtained from gWAT of lean male 8 week old C57BL/6 mice. n = 4. (G) mRNA levels of Cluster 2 genes in same sorted populations shown in (F). n = 4. (H) mRNA levels of Cluster 3 genes in same sorted populations shown in (F). n = 4. * in all graphs denote p<0.05 by student's t-test in comparisons to the other populations. Bars represent mean +SEM.

DOI: https://doi.org/10.7554/eLife.39636.010

The following source data is available for figure 2:

**Source data 1.** Complete list of differentially expressed genes (k-means = 3).

DOI: https://doi.org/10.7554/eLife.39636.011

adipocyte differentiation at a high efficiency, while very few adipocytes emerged in the other two PDGFRβ+ subpopulations or within cultures containing all PDGFRβ+ cells from gWAT (*Figure 3*). FIPs appeared to possess some latent capacity to undergo adipogenesis. Confluent cultures of LY6C + PDGFRβ+ cells stimulated with a more commonly used hormonal adipogenic cocktail (dexamethasone, IBMX, insulin, and PPARγ agonist, Rosiglitazone) underwent to adipocyte differentiation to some degree (*Figure 3—figure supplement 2*). Despite this strong adipogenic stimulus, LY6C+ PDGFRβ+ cells still did not differentiate to the same extent as LY6C- CD9- PDGFRβ+ cells stimulated with insulin alone (see *Figure 3*). We also assessed the ability of APCs and FIPs to undergo adipocyte differentiation in vivo. We transplanted 80,000 cells into the remnant subcutaneous WAT depots of *Adipoq*-Cre; *Pparg*^loxP/loxP animals, a well-described model of lipodystrophy (*Figure 3—figure supplement 3A*) (*Wang et al., 2013a*). 3 weeks following cell transplantation, the WAT depots all four animals injected with LY6C- CD9- PDGFRβ+ cells contain numerous clusters of lipid-laden fat cells (*Figure 3—figure supplement 3B*). The contralateral depots of the same animals injected with LY6C+ PDGFRβ+ cells, or matrigel alone, remained devoid of adipocytes (*Figure 3—figure supplement 3C,D*). Collectively, these data indicate that LY6C- CD9- PDGFRβ+ cells are highly adipogenic functional gonadal white adipocyte precursors, while LY6C+ PDGFRβ+ cells are largely refractory to adipogenic stimuli.

Several studies have defined APCs from gonadal WAT as SCA-1+ CD34+ CD24± cells that also express PDGFRα (*Berry and Rodeheffer, 2013*; *Jeffery et al., 2015*; *Lee et al., 2012*; *Rodeheffer et al., 2008*). In fact, most studies of gonadal WAT APCs isolate these cells on the basis of these markers. Additionally, recent studies identified CD38 as a marker of committed preadipocytes (*Carrière et al., 2017*). The scRNA-seq analysis and follow-up qPCR analyses of isolated subpopulations revealed that all three PDGFRβ+ subpopulations indeed expressed *Pdgfra*, *Ly6a* (SCA-1), and *Cd34*; however, the mRNA levels of *Ly6a* and *Cd34* are actually lower in LY6C- CD9- PDGFRβ+ APCs than in LY6C+ PDGFRβ+ cells (FIPs) (*Figure 3—figure supplement 1F,G*). As expected, all three subpopulations expressed *Pdgfrb*; however, mRNA levels of *Pdgfrb* were quantitatively lower in FIPs than in the APCs and MLCs. qPCR analysis indicated that levels of *Cd24a* were low in all three PDGFRβ+ subpopulations. *Cd38* was present predominately in LY6C- CD9- PDGFRβ + cells, consistent with the notion that CD38 identifies APCs from this depot (*Carrière et al., 2017*) (*Figure 3—figure supplement 1F,G*). Flow cytometry analyses revealed similar patterns of surface protein expression in these subpopulations (*Figure 3—figure supplement 1H*). Collectively, these data reveal the selection of gonadal WAT SVF cells on the basis of SCA-1/CD34 yields functionally heterogeneous cell populations, and perhaps biases against the selection of LY6C- CD9- PDGFRβ+ APCs.

Recently, Burl et al. reported scRNA-seq profiles of adipose SVF cells, creating a cellular atlas of potential adipocyte precursor populations (perivascular and non-perivascular) (*Burl et al., 2018*). Notably, the authors identified two prominent populations within the gonadal WAT depot, termed adipose stem cell (ASC) 1 and ASC 2. Moreover, they identified two additional smaller ASC subpopulations that were considered 'differentiating' ASCs and 'proliferating' ASCs. The identified populations were not isolated and explored functionally in their study; however, a comparison of the molecular profiles strongly suggests that ASC 1 defined by the authors bears close resemblance to APC population defined in our study, while the ASC 2 population bears close resemblance to the FIPs discovered here (*Figure 3—figure supplement 1I*). Markers of the differentiated/proliferative

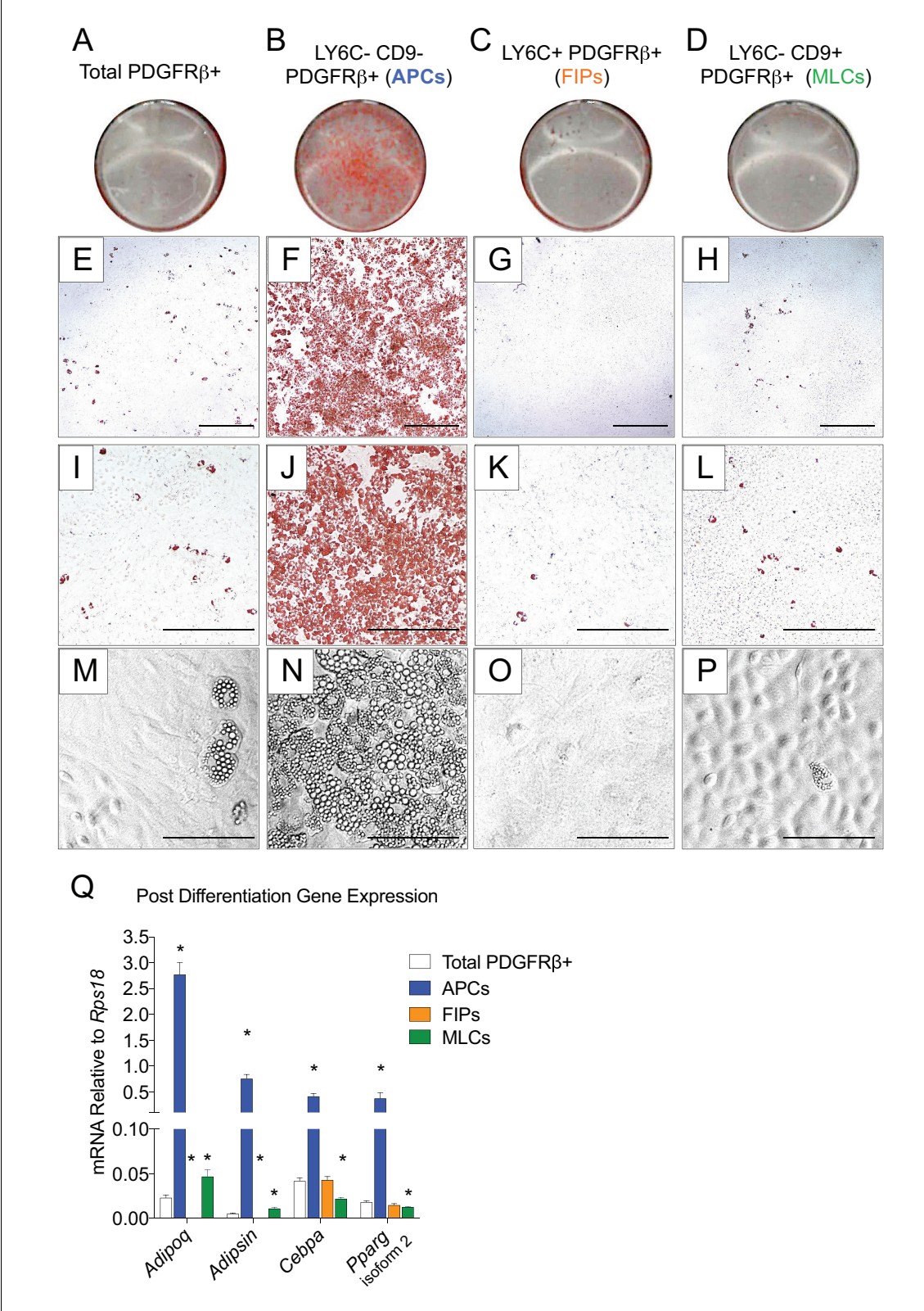

**Figure 3.** LY6C- CD9- PDGFRβ+ cells (APCs) are functional gonadal white adipocyte precursors. (**A**) Photograph of Oil Red O (ORO) stained gWAT-derived PDGFRβ+ cells maintained for 8 days in growth media (2% FBS and ITS supplement). (**B**) Photograph of ORO stained LY6C- CD9- PDGFRβ+ cells maintained for 8 days in growth media. (**C**) Photograph of ORO stained LY6C+ PDGFRβ+ cells maintained for 8 days in growth media. (**D**) Photograph of ORO stained LY6C- CD9+ PDGFRβ+ cells maintained for 8 days in growth media. (**E**) Brightfield image of the culture shown in A. Scale

*Figure 3 continued on next page*

*Figure 3 continued*

bar = 400 µm. (F) Brightfield image of the culture shown in B. Scale bar = 400 µm. (G) Brightfield image of the culture shown in C. Scale bar = 400 µm. (H) Brightfield image of the culture shown in D. Scale bar = 400 µm. (I) Brightfield image of the culture shown in A. Scale bar = 200 µm. (J) Brightfield image of the culture shown in B. Scale bar = 200 µm. (K) Brightfield image of the culture shown in C. Scale bar = 200 µm. (L) Brightfield image of the culture shown in D. Scale bar = 200 µm. (M) Brightfield image of unstained PDGFRβ+ cells maintained for 8 days in growth media. Scale bar = 100 µm. (N) Brightfield image of unstained LY6C- CD9- PDGFRβ+ cells maintained for 8 days in growth media. Scale bar = 100 µm. (O) Brightfield image of unstained LY6C+ PDGFRβ+ cells maintained for 8 days in growth media. Scale bar = 100 µm. (P) Brightfield image of unstained LY6C- CD9+ PDGFRβ+ cells maintained for 8 days in growth media. Scale bar = 100 µm. (Q) mRNA levels of adipocyte-selective genes in total PDGFRβ+ cells, APCs, FIPs, and MLCs, after 8 days of culture in growth media. * denotes p<0.05 by student's t-test in comparisons to total PDGFRβ+ cells. Bars represent mean +SEM. n = 4–7. All photographs/images are representative of multiple experiments/repetitions (See *Supplementary file 1*).

DOI: https://doi.org/10.7554/eLife.39636.012

The following figure supplements are available for figure 3:

**Figure supplement 1.** Expression of common adipocyte stem cell markers in APCs, FIPs, and MLCs, isolated from gonadal WAT of adult male mice.

DOI: https://doi.org/10.7554/eLife.39636.013

**Figure supplement 2.** FIPs undergo adipocyte differentiation in the presence of dexamethasone, IBMX, insulin, and rosiglitazone.

DOI: https://doi.org/10.7554/eLife.39636.014

**Figure supplement 3.** Visceral APCs undergo adipocyte differentiation upon transplantation into lipodystrophic mice.

DOI: https://doi.org/10.7554/eLife.39636.015

**Figure supplement 4.** Gonadal PDGFRβ+ *Zfp423*GFP-High cells enrich for markers of committed preadipocytes.

DOI: https://doi.org/10.7554/eLife.39636.016

ASCs aligned closely to the committed PDGFRβ+ preadipocyte depicted in *Figure 1B*. Taken together, our data here suggest a refined strategy to isolate functional white adipocyte precursors from visceral WAT of adult mice.

Our prior studies of *Zfp423*GFP reporter mice indicated that gonadal WAT PDGFRβ+ cells expressing GFP are enriched in the expression of *Pparg* and are highly adipogenic in vitro (*Gupta et al., 2012*; *Vishvanath et al., 2016*). Additional studies by others indicated that this reporter captures committed preadipocytes within the skeletal bone marrow microenvironment (*Ambrosi et al., 2017*). Endogenous *Zfp423* mRNA levels were found in all PDGFRβ+ subpopulations, albeit at highest levels in APCs. (*Figure 3—figure supplement 4A*). We re-examined *Zfp423*GFP-High and *Zfp423*GFP-Low PDGFRβ+ cells isolated from gWAT (*Figure 3—figure supplement 4B*), asking whether these labeled cells captured by this reporter allele enriched for any of the Cluster markers identified by scRNA-seq. Consistent with our prior studies, *Zfp423*GFP-High PDGFRβ+ cells were enriched in the expression of *Pparg* isoforms when compared to *Zfp423*GFP-Low PDGFRβ+ cells (*Figure 3—figure supplement 4C*). Further gene expression analysis of the top cluster gene markers revealed that *Zfp423*GFP-High cells were enriched in the expression of the genes that define the APC clusters, but not FIPs or MLCs (*Figure 3—figure supplement 4D–G*). In particular, *Zfp423*GFP-High PDGFRβ+ cells were enriched in the expression of genes that delineate the more committed preadipocytes cluster (Cluster 1B) identified by scRNA-seq (*Figure 3—figure supplement 4E*). Taken all together, these data indicate that endogenous *Zfp423* mRNA expression is not confined exclusively to the APC subpopulation of PDGFRβ+ cells in gWAT; however, *Zfp423*GFP reporter mice represent a genetic tool to localize and enrich for committed preadipocytes from this depot.

## Functionally distinct stromal populations from visceral, but not subcutaneous, WAT depots can be revealed on the basis of LY6C and CD9 expression

Transcriptional programs of white adipocyte precursors are depot- and sex dependent (*Macotela et al., 2012*). Thus, we asked whether similar functional heterogeneity exists amongst PDGFRβ+ cells within various WAT depots, and whether functionally distinct subpopulations could be selected for using the same FACS strategy described above. Indeed, the same three populations can be observed within the mesenteric and retroperitoneal depots of adult male mice, with LY6C-CD9- PDGFRβ+ cells representing the highly adipogenic subpopulation (*Figure 4A–H*). We also examined LY6C expression within PDGFRβ+ SVF cells obtained from the inguinal and anterior subcutaneous WAT depots. We previously demonstrated that the total pool of PDGFRβ+ cells from

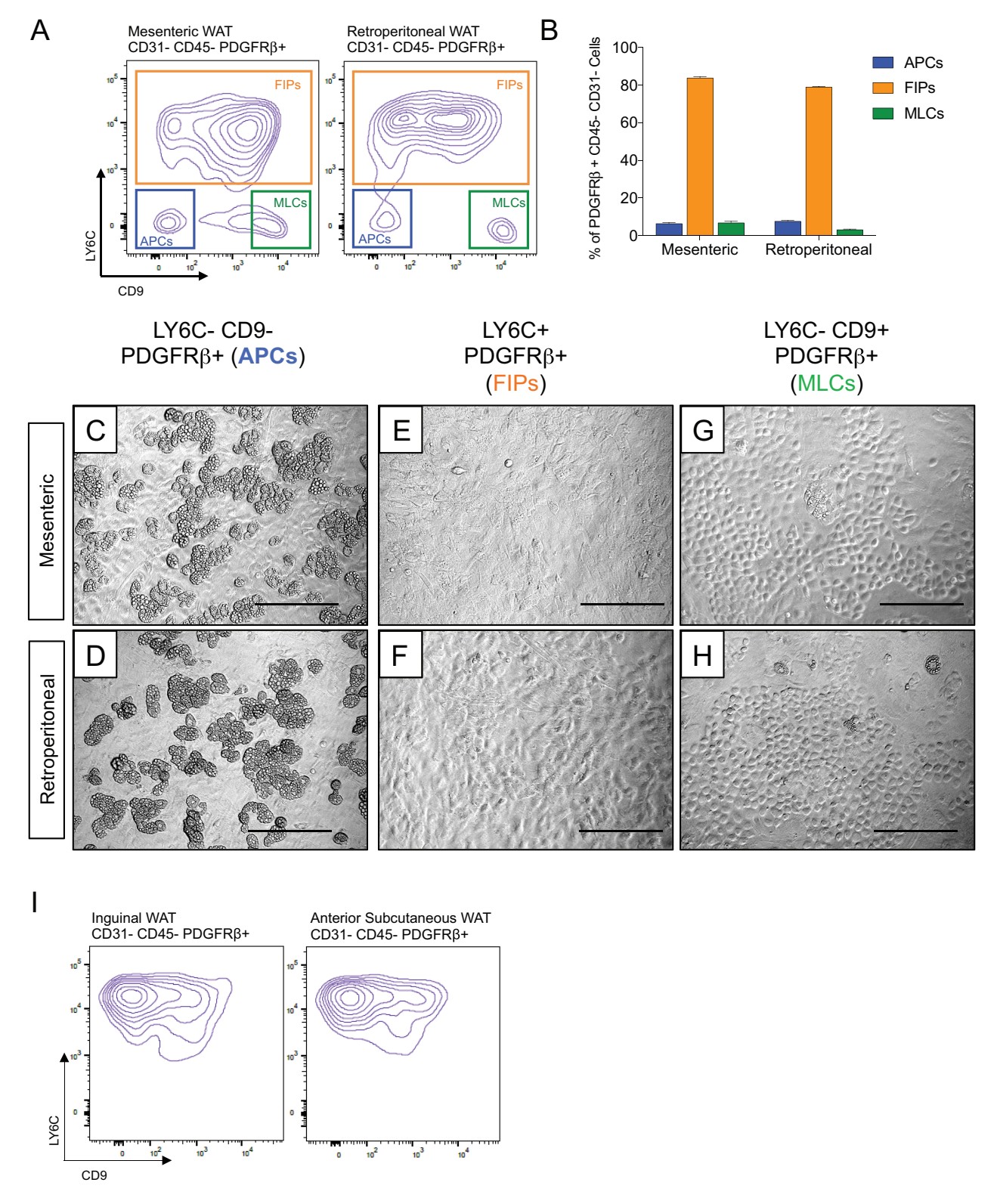

**Figure 4.** Functionally distinct stromal populations from visceral, but not subcutaneous, WAT depots can be revealed on the basis of LY6C and CD9 expression. (**A**) Fluorescence-activated cell sorting (FACS) gating strategy to isolate indicated PDGFRβ+ CD31- CD45- subpopulations from mesenteric and retroperitoneal WAT. (**B**) Frequency of APCs, FIPs, and MLCs in mesenteric and retroperitoneal WAT isolated from lean male 8 week old C57BL/6 mice. Frequencies were quantified based on the gating strategy shown in (**A**). n = 6. Bars represent mean +SEM. (**C**) Brightfield image of LY6C- CD9-
*Figure 4 continued on next page*

*Figure 4 continued*

PDGFRβ+ (APCs) cells from mesenteric WAT maintained for 8 days in growth media. Scale bar = 200 μm. (**D**) Brightfield image of LY6C+ PDGFRβ+ (FIPs) cells from mesenteric WAT maintained for 8 days in growth media. Scale bar = 200 μm. (**E**) Brightfield image of LY6C- CD9+ PDGFRβ+ (MLCs) cells from mesenteric WAT maintained for 8 days in growth media. Scale bar = 200 μm. (**F**) Brightfield image of LY6C- CD9- PDGFRβ+ (APCs) cells from retroperitoneal WAT maintained for 8 days in growth media. Scale bar = 200 μm. (**G**) Brightfield image of LY6C+ PDGFRβ+ (FIPs) cells from retroperitoneal WAT maintained for 8 days in growth media. Scale bar = 200 μm. (**H**) Brightfield image of LY6C- CD9+ PDGFRβ+ (MLCs) cells from retroperitoneal WAT maintained for 8 days in growth media. Scale bar = 200 μm. (**I**) Flow cytometry plot of LY6C and CD9 expression in CD31- CD45- PDGFRβ+ cells isolated from inguinal WAT and anterior subcutaneous WAT.

DOI: https://doi.org/10.7554/eLife.39636.017

The following figure supplement is available for figure 4:

**Figure supplement 1.** APCs and FIPs can be isolated from gonadal WAT of female mice.

DOI: https://doi.org/10.7554/eLife.39636.018

inguinal WAT is very highly adipogenic in vitro (*Shao et al., 2018*); however, remarkably, all PDGFRβ+ cells within the inguinal and anterior subcutaneous WAT depots expressed LY6C (*Figure 4I*). These data suggest that if heterogeneity exists amongst PDGFRβ+ cells in these subcutaneous depots, subpopulations could not be discriminated on the basis of LY6C expression. Therefore, functionally distinct perivascular cell subpopulations from visceral, but not subcutaneous, WAT depots can be revealed on the basis of LY6C and CD9 expression.

We also asked whether visceral WAT in female mice contains APCs and FIPs, bearing similar molecular and functional properties. Within the SVF of peri-ovarian WAT, the same three distinct subpopulations of PDGFRβ+ cells can be discriminated, with FIPs being the predominant population (*Figure 4—figure supplement 1A,B*). Importantly, gene expression analysis by qPCR confirmed that LY6C- CD9- PDGFRβ+ cells were enriched in the expression of genes that defined the epididymal WAT APC population (Cluster 1) (*Figure 4—figure supplement 1C,F*), including *Pparg* isoform 2. LY6C+ PDGFRβ+ cells enriched for the mRNAs that initially defined the epididymal WAT FIPs (Cluster 2) (*Figure 4—figure supplement 1D,G,H*), and LY6C- CD9+ PDGFRβ+ cells expressed mesothelial/epithelial markers (*Figure 4—figure supplement 1E*). Moreover, LY6C- CD9- PDGFRβ+ cells from peri-ovarian WAT are functional adipocyte precursors; these cells, but neither FIPs nor MLCs, differentiate spontaneously upon reaching confluence in culture (*Figure 4—figure supplement 1I–K*). Collectively, these data provide evidence that functional APCs from both male and female visceral WAT can be isolated through this cell sorting strategy.

## Visceral LY6C+ PDGFRβ+ cells are anti-adipogenic and appear molecularly distinct from inguinal WAT Aregs

It is notable that very little spontaneous adipocyte differentiation occurs in cultures containing the total pool of visceral adipose PDGFRβ+ cells (*Figure 3A,E,I,M,Q*), despite the presence of numerous APCs within this population. This suggested that perhaps the presence of FIPs within these cultures influenced the differentiation capacity of neighboring APCs in vitro. Therefore, we also tested the impact of conditioned media from cultured FIPs on the differentiation capacity of APCs residing in parallel cultures. Remarkably, APCs exposed to conditioned media from FIPs, but not from parallel cultures of APCs, expressed lower levels of *Pparg* (*Figure 5A*). Moreover, APCs exposed to conditioned media from FIPs lost a significant degree of adipogenic capacity (*Figure 5B,C,E*). Conditioned media from cultures of MLCs had only a slight inhibitory effect on the terminal differentiation of APCs (*Figure 5D,E*). Collectively, these data not only suggest that FIPs lack significant adipogenic capacity, but highlight the notion that these cells can actually be anti-adipogenic.

Recently, Schwalie et al. identified anti-adipogenic stromal cells within the inguinal WAT of mice (*Schwalie et al., 2018*). These cells, termed Aregs, are defined, in part, by the expression of CD142 and ABCG1 and exhibit perivascular localization. From our scRNA-seq dataset, we observed that *F3* expression (encoding CD142) is detected in all PDGFRβ+ clusters of gonadal WAT, albeit not enriched in FIPs (*Figure 5F*). *Abcg1* expression was not detected by the sequencing analysis in any population. We also examined the levels of mRNA for these two markers directly by quantitative PCR analysis. Consistent with the sequencing data, neither marker was enriched in FIPs (*Figure 5G*). We also examined additional genes (23 in total) whose expression defines Aregs, as identified by Schwalie et al, by assessing their expression level within our scRNA-seq dataset (*Figure 5H*). Levels

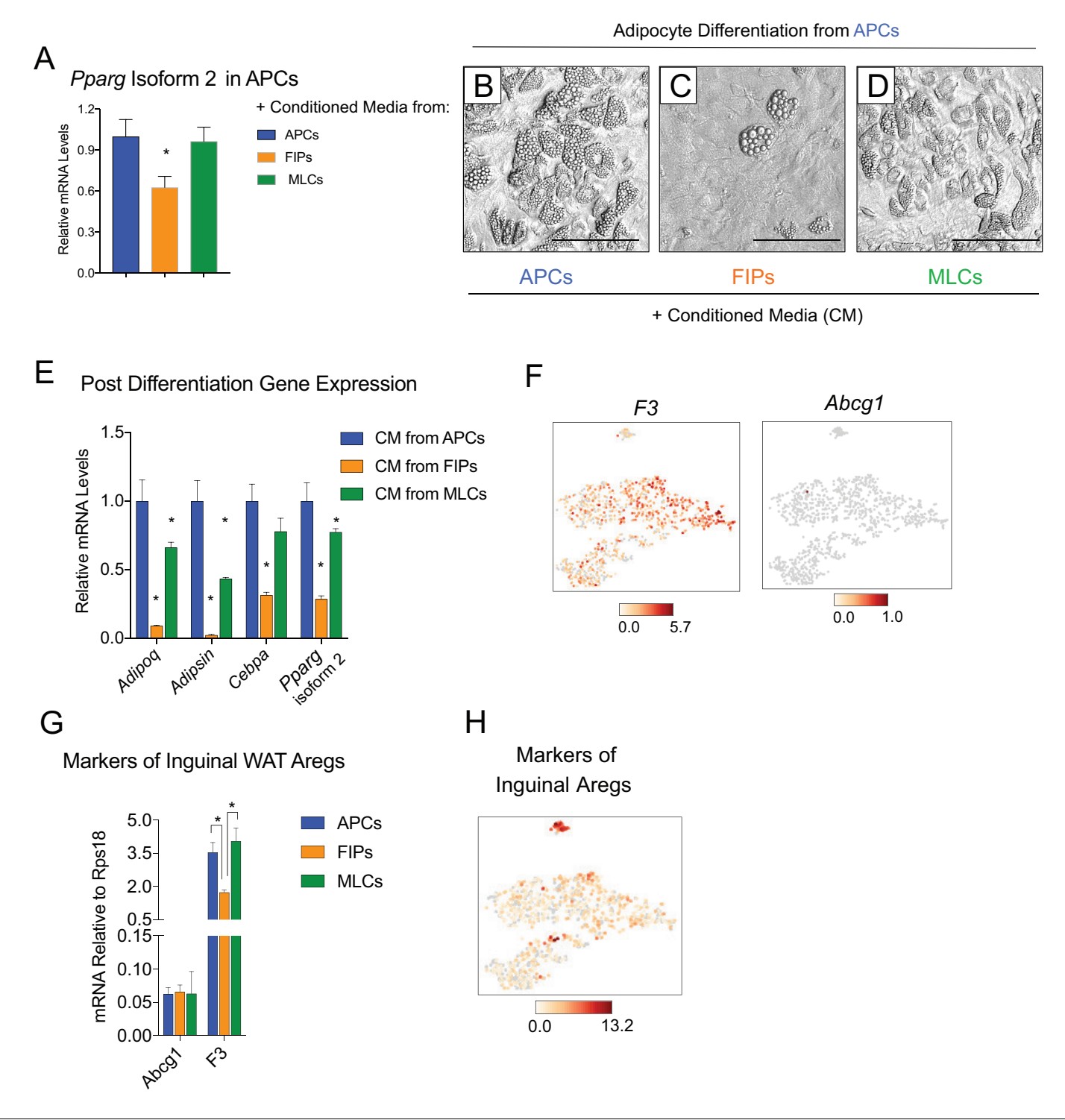

**Figure 5.** FIPs inhibit adipocyte differentiation from APCs. (A) *Pparg* isoform two expression in cultured APCs maintained for 3 days in conditioned media from either APCs, FIPs, or MLCs. n = 4. *denotes p<0.05 by student's t-test in comparisons to data represented in blue bars. Bars represent mean +SEM. (B) Brightfield image of APCs after 8 days of culture in conditioned media from parallel cultures of APCs. Scale bar = 100 μm for B-D. (C) Brightfield image of APCs after 8 days of culture in conditioned media from parallel cultures of FIPs. (D) Brightfield image of APCs after 8 days of culture in conditioned media from parallel cultures of MLCs. (E) mRNA levels of adipocyte-selective genes within cultures shown in (B–D). n = 3. * denotes p<0.05 by student's t-test in comparisons to data represented in blue bars. Bars represent mean +SEM. (F) Distribution of *Abcg1* and *F3* expression within *tSNE* plot from *Figure 1B*. (G) mRNA levels of *Abcg1* and *F3* in APCs, FIPs, and MLCs isolated from lean 8 week old male mice. *

*Figure 5 continued on next page*

*Figure 5 continued*
denotes p<0.05 by student's t-test. Bars represent mean +SEM. (**H**) tSNE-plot highlighting the potential relationship between APCs, FIPs, and MLCs, and iguinal WAT Aregs identified by Schwalie et al. (***Schwalie et al., 2018***). The top-23 Areg-selective genes identified by Schwalie et al were input into Cell Loupe Browser. Color intensities represent the sum of the $Log_2$ expression values of the Areg selective gene list within the single cell RNA-sequencing dataset of gWAT from ***Figure 1B***.
DOI: https://doi.org/10.7554/eLife.39636.019

of transcripts corresponding to a number of these genes were detectable, albeit at low levels. Notably, there was no selective enrichment of the broader set of Areg markers within FIPs or APCs. As such, despite some shared functional similarities, inguinal adipose Aregs and the gonadal adipose FIPs described here appear molecularly distinct.

## Visceral LY6C+ PDGFRβ+ cells are fibrogenic and exert a functional pro-inflammatory phenotype

As described above, GSEA of scRNA-seq profiles also identified a gene expression profile suggestive of active TGFβ signaling within Cluster 2 cells (***Table 3***). Indeed the expression of major collagens (*Col1a1* and *Col3a1*) and some of the assayed genes associated with extracellular matrix accumulation were enriched in freshly isolated LY6C+ PDGFRβ+ cells compared to the other PDGFRβ+ subpopulations (***Figure 6—figure supplement 1A***). In vitro, cultured FIPs and APCs were both responsive to treatment with recombinant TGFβ; however, the expression of collagens examined remained higher and/or was further induced in FIPs (***Figure 6—figure supplement 1B***). These data indicate that LY6C+ PDGFRβ+ FIPs exhibit a phenotype characteristic of fibrogenic cells.

The most striking result from GSEA was the enrichment of pathways related to active 'Tnfα signaling' and 'inflammatory response' in FIPs (***Table 3***). Remarkably, FIPs exhibited a robust inflammatory gene expression signature following acute exposure to pro-inflammatory molecules. Lipopolysaccharide (LPS) treatment induced inflammatory cytokine gene expression in both APCs and FIPs; however, the response was more robust in the latter population (***Figure 6A***). The differential response to TNFα treatment was the most striking; FIPs, but not APCs, activate the expression of several pro-inflammatory cytokines under these conditions (***Figure 6B***). These fibro-inflammatory cells displayed increased gene expression of numerous cytokines involved in the recruitment of leukocytes and the activation of immune cells. This suggested that FIPs have the potential to activate macrophages through cytokine production. To test this, we treated cultured bone marrow derived macrophages with conditioned media from LPS-treated FIPs, APCs, and MLCs (***Figure 6C***). Macrophage cultures exposed to conditioned media from LPS-treated FIPs had the most robust induction of pro-inflammatory genes, including *Tnfα*, *Il1β*, and *Il6* (***Figure 6D***). These data highlight the potential for FIPs to exert a functional pro-inflammatory phenotype.

## The frequencies and gene expression profiles of APCs and FIPs are differentially regulated in association with high-fat diet feeding

In the setting of caloric excess, adipose tissue undergoes a dramatic remodeling as it expands to meet increased demands for energy storage. Shortly after the onset of high-fat diet (HFD) feeding, adipose tissue inflammation occurs (***Hill et al., 2014***; ***Xu et al., 2003***). After 4–5 weeks of HFD feeding (60% kcal from fat), newly formed visceral adipocytes emerging from the PDGFRβ+ lineage begin to appear (***Vishvanath et al., 2016***). We asked if the frequency of FIPs and APCs were altered during the course of HFD feeding. Four weeks of HFD feeding did not appear to dramatically alter the absolute number of PDGFRβ+ cells present in gWAT; however, the ratio of FIPs to APCs begins to increase by as early as one week of HFD feeding (***Figure 7A***). We also analyzed BrdU incorporation into the mural cell populations during one week of HFD feeding. FIPs and the MLCs displayed the greatest BrdU incorporation (***Figure 7B,C***). BrdU incorporation into APCs was significantly lower than observed in the FIPs (***Figure 7B,C***). These data indicate that frequencies of APCs and FIPs are differentially regulated in vivo in association with high-fat diet feeding, with FIPs exhibiting a relatively higher degree of cell proliferation under these conditions.

The change in frequency of FIPs and APCs during HFD feeding prompted us to examine if their defining gene expression programs were altered under these conditions. One month of HFD feeding

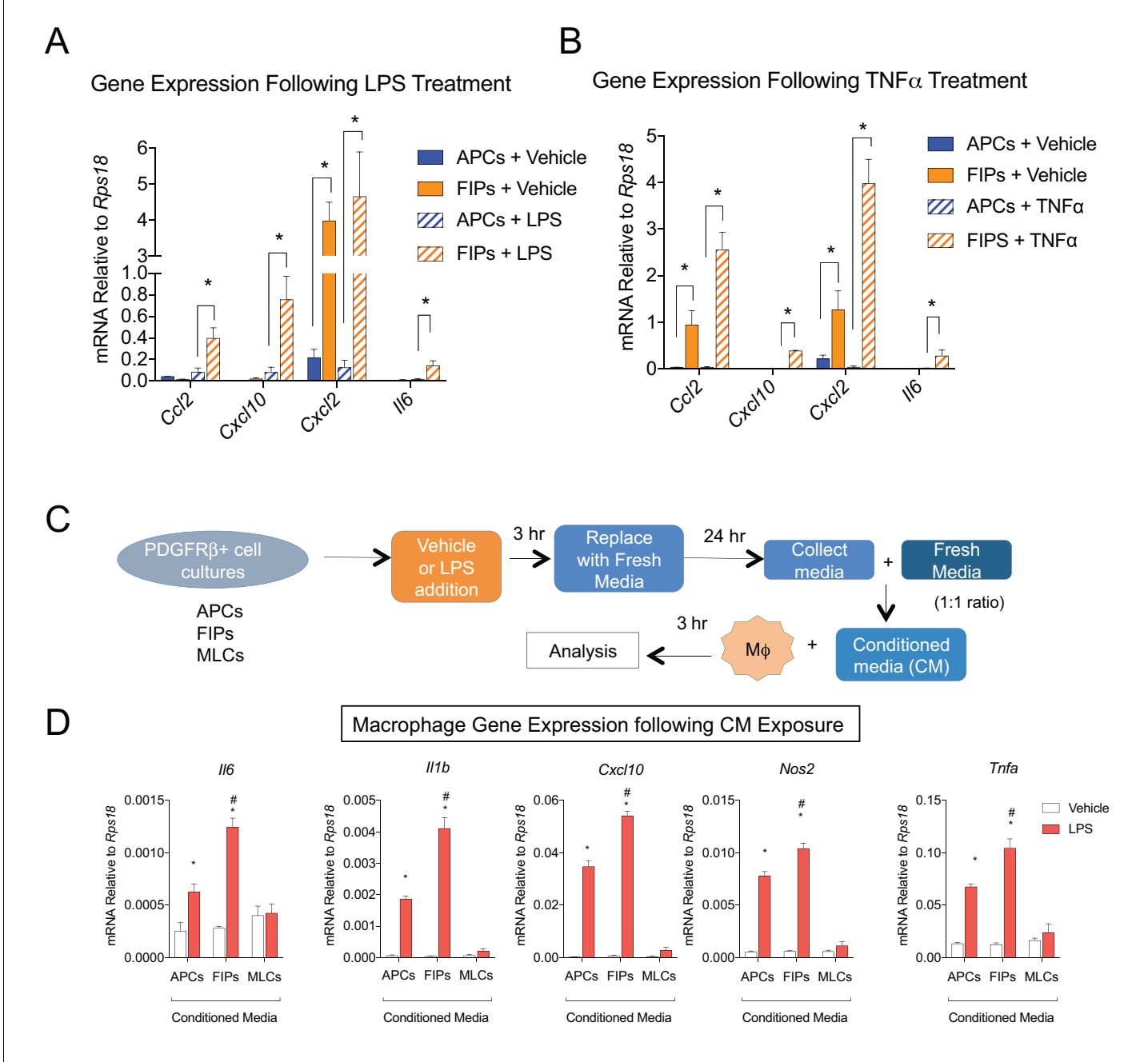

**Figure 6.** LY6C+ PDGFRβ+ cells (FIPs) exhibit a functional pro-inflammatory phenotype. (**A**) mRNA levels of indicated cytokines in cultures of APCs and FIPs treated with vehicle (PBS) or LPS (100 ng/ml) for 3 hr. * denotes p<0.05 by student's t-test. Bars represent mean +SEM. n = 4. (**B**) mRNA levels of indicated cytokines in cultures of APCs and FIPs treated with vehicle (PBS) or TNFα (20 ng/ml) for 3 hr. * denotes p<0.05 by student's t-test. Bars represent mean +SEM. n = 4. (**C**) Schematic depicting the treatment of bone marrow derived macrophages (MΦ) with conditioned media (CM) from LPS-treated APCs, FIPs and MLCs. (**D**) mRNA levels of select markers of activated macrophages in macrophage cultures exposed to conditioned media. n = 4. * denotes p<0.05 comparing vehicle vs. LPS. # denotes p<0.05 comparing LPS-treated FIPs vs. LPS-treated APCs. Bars represent mean +SEM.

DOI: https://doi.org/10.7554/eLife.39636.020

The following figure supplement is available for figure 6:

**Figure supplement 1.** LY6C+ PDGFRβ+ cells (FIPs) exhibit a fibrogenic phenotype.

DOI: https://doi.org/10.7554/eLife.39636.021

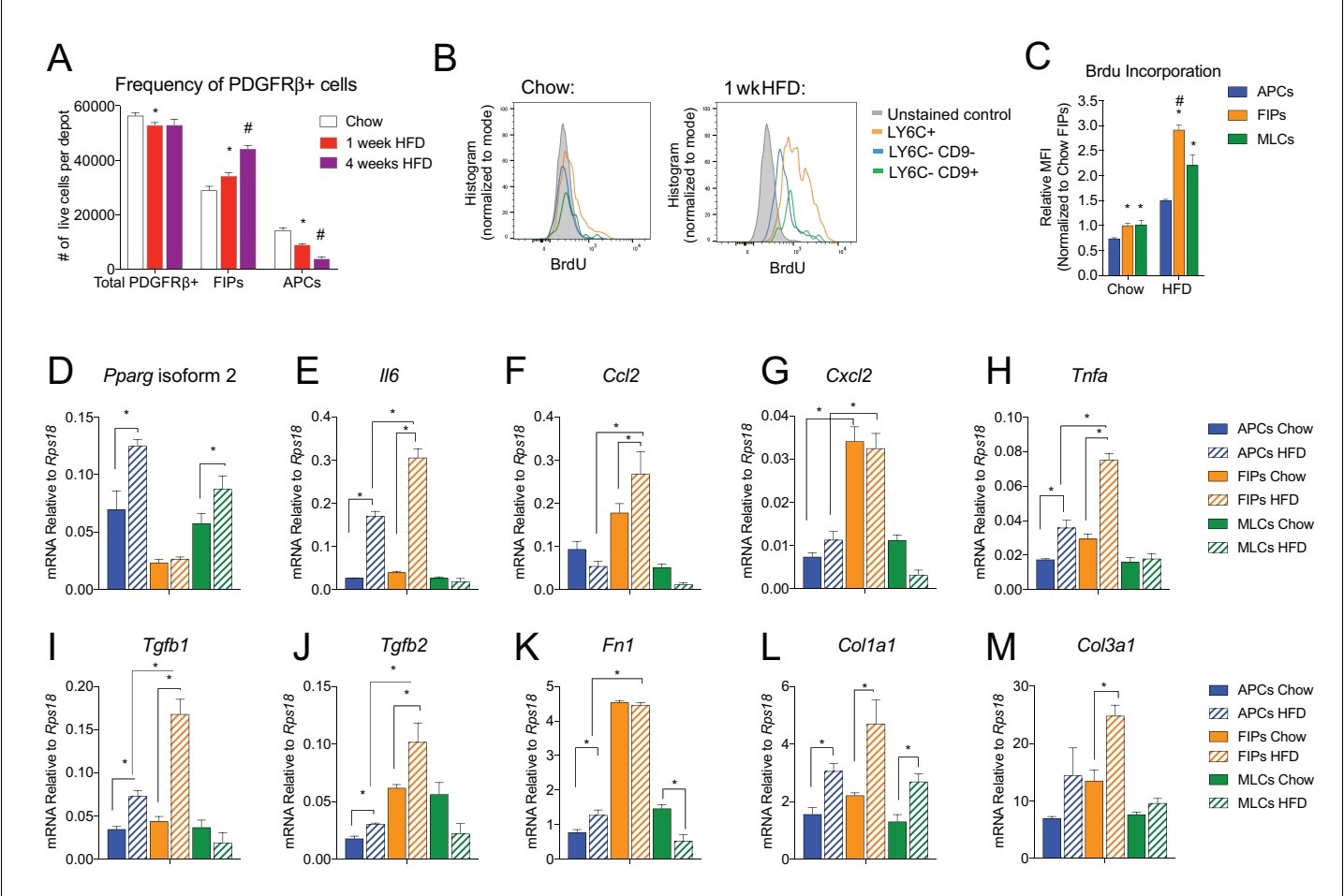

**Figure 7.** The frequencies and gene expression profiles of APCs and FIPs are differentially regulated in association with high-fat diet feeding. (A) Frequency of total PDGFRβ+ cells, FIPs, and APCs in gonadal WAT isolated from chow-fed mice, mice fed high fat diet (HFD) for 1 week, or mice fed HFD for 4 weeks. n = 4. * denotes p<0.05 by student's t-test in comparison to white bars. # denotes p<0.05 by student's t-test in comparison to red or white bars. Bars represent mean +SEM. (B) Histograms depicting BrdU incorporation into APCs, FIPs, and MLCs after 1 week of chow or HFD feeding. (C) Relative median fluorescence intensity (MFI) corresponding to histograms shown in (B). n = 4. * denotes p<0.05 by student's t-test in comparison to corresponding data from APCs. # denotes p<0.05 by student's t-test in comparison to corresponding data from APCs and MLCs. (D) *Pparg* isoform two expression in freshly isolated APCs, FIPs, and MLCs, from gWAT of chow or 4 week HFD fed mice. n = 4. (E) *Il6* expression in same cell populations shown in (D). (F) *Ccl2* expression in same cell populations shown in (D). (G) *Cxcl2* expression in same cell populations shown in (D). (H) *Tnfa* expression in same cell populations shown in (D). (I) *Tgfb1* expression in same cell populations shown in (D). (J) *Tgfb2* expression in same cell populations shown in (D). (K) *Fn1* expression in same cell populations shown in (D). (L) *Col1a1* expression in same cell populations shown in (D). (M) *Col3a1* expression in same cell populations shown in (D). * in panels D-M denote p<0.05 by student's t-test. All bars represent mean +SEM.
DOI: https://doi.org/10.7554/eLife.39636.022

lead to a significant elevation in mRNA levels of *Pparg* isoform two expression in APCs, with a smaller increase occurring in MLCs (*Figure 7D*). *Pparg* isoform two expression was not elevated in FIPs, consistent with their apparent lack of adipogenic potential (*Figure 7D*). mRNA levels of pro-inflammatory cytokines and extracellular matrix components were further induced and/or remained more abundant in FIPs than in APCs or MLCs (*Figure 7E–M*). Interestingly, APCs activated the expression of some of these genes (e.g. *Il6*, *Tnfa*, *Col1a1*, *Col3a1*) during HFD feeding. These data are consistent with the in vitro analyses highlighting the potential of APCs to trigger some degree of an inflammatory response in pro-inflammatory stimuli (see *Figure 6*). These data reveal that PDGFRβ + subpopulations exhibit unique transcriptional responses to HFD feeding; however, these data also suggest that APCs have some capacity to adopt characteristics of FIPs in vivo.

## NR4A nuclear receptors regulate the pro-inflammatory phenotype of PDGFRβ+ cells

We sought to gain insight into the potential transcriptional mechanisms regulating the pro-inflammatory and adipogenic phenotypes of PDGFRβ+ perivascular cells. A number of transcription factors were differentially expressed between FIPs and APCs; however, it was notable that the expression of all three members of the *Nr4a* family of nuclear hormone receptors was significantly enriched in the FIPs cluster (*Figure 8A*). Gene expression analysis by qPCR of the isolated populations confirmed the significant enrichment of *Nr4a1*, *Nr4a2*, and *Nr4a3* in FIPs isolated from chow-fed mice, with relatively lower expression in the APCs and MLCs (*Figure 8B*).

Members of the NR4A family, including NR4A1 (NUR77), NR4A2 (NURR1), and NR4A3 (NOR1), have been implicated in the regulation of inflammation; however, their exact impact on inflammatory signaling appears cell-type specific (*Rodríguez-Calvo et al., 2017*). Following 4 weeks of HFD-feeding, the expression of *Nr4a* family members remained significantly enriched in FIPs (*Figure 8C*). In vitro, the expression of all *Nr4a* family members in FIPs was increased following exposure to recombinant TNFα (*Figure 8D*). Therefore, we assessed the consequences of retroviral-mediated overexpression of individual *Nr4a* family members on the inflammatory response of FIPs (*Figure 8E–G*). Overexpression of *Nr4a1*, *Nr4a2*, or *Nr4a3*, attenuated the pro-inflammatory response to TNFα (*Figure 8H–K*). Moreover, we assessed the impact of retroviral-mediated knockdown of *Nr4a1* on the inflammatory response in FIPs. Knockdown of *Nr4a1* using three independent shRNAs led to an exaggerated response of FIPs to TNFα treatment (*Figure 8L–P*). These gain- and loss of function studies suggest that FIPs activate *Nr4a* family members in response to pro-inflammatory stimuli, perhaps as a means to counter-regulate a sustained cellular inflammatory response. These data provide proof of concept that FIPs may be utilized as a tool to identify additional regulators of inflammatory signaling pathways.

## Discussion

Visceral adipose tissue dysfunction in obesity is driven, at least in part, by chronic tissue inflammation, collagen deposition, and a loss of adipocyte precursor activity. WAT remodeling involves substantial qualitative and quantitative changes to the composition of the stromal compartment of the tissue; however, the functional heterogeneity of WAT stromal-vascular fraction has remained poorly defined and tools to isolate and study distinct subpopulations have been lacking. Here, we unveil functionally distinct PDGFRβ+ stromal cell subpopulations in visceral WAT (*Figure 9*). Importantly, we have developed strategies to prospectively isolate these distinct populations using commercially available antibodies.

Functional analyses indicate that a relatively large subpopulation of PDGFRβ+ perivascular cells in visceral gonadal WAT exert fibrogenic and pro-inflammatory phenotypes. These cells, termed here as 'FIPs,' lack adipogenic capacity in vitro but instead exhibit a fibrogenic phenotype. FIPs are physiologically regulated; the frequency of these cells increases upon HFD feeding. Clement and colleagues previously reported that fibrogenic cells residing in WAT could be identified by the expression of CD9 and PDGFRα (*Marcelin et al., 2017*). Indeed, FIPs express CD9 and PDGFRα; however, both CD9 and PDGFRα are also expressed in at least a subpopulation of mesothelial cells isolated from visceral WAT. As such, the selection of FIPs on the basis of LY6c and PDGFRβ expression (LY6C+ PDGFRβ+) represents a strategy to prospectively and specifically isolate FIPs from mouse gonadal WAT. Importantly, our data reveal a number of previously unappreciated ways in which perivascular cells may impact WAT remodeling (*Figure 9*). The LY6C+ PDGFRβ+ cells described here have the capacity to inhibit adipocyte differentiation from APCs through the release of secreted factors. The presence of highly anti-adipogenic stromal cells within the total PDGFRβ+ population may explain the apparent lack of adipogenic capacity that crude/unpurified visceral PDGFRβ+ cultures possess vitro, despite the presence of APCs. It is notable that visceral adipose FIPs appear distinct from the recently identified Aregs of inguinal WAT. Stromal cell heterogeneity may be depot-specific, with different depots utilizing distinct cell types to control their function and plasticity. The exact identities of the secreted factors and mechanisms that mediate the anti-adipogenic activity of FIPs and Aregs are still unknown. Importantly, whether FIPs and Aregs act to suppress/restrain adipocyte hyperplasia under physiological settings in vivo needs to be further explored.

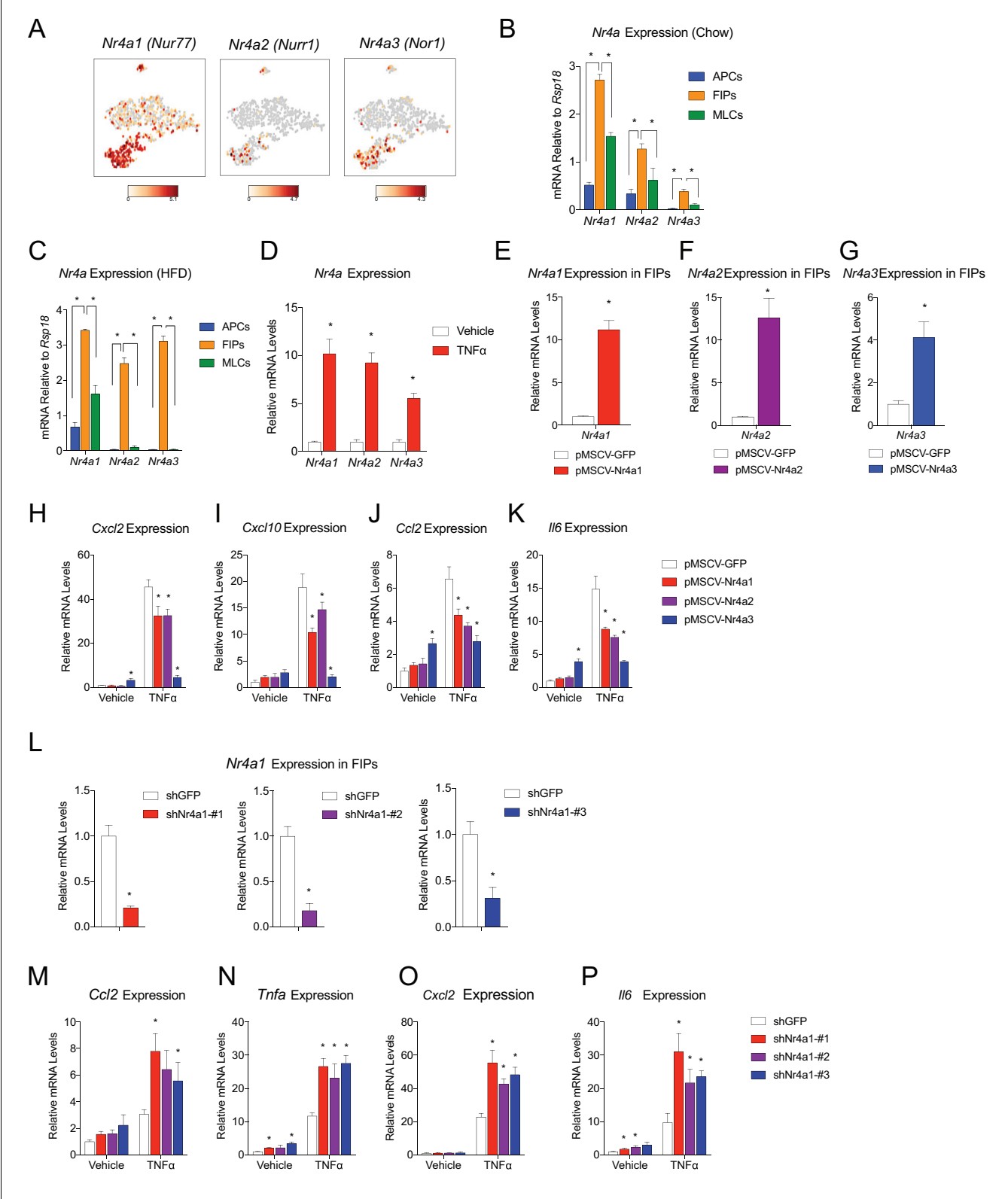

**Figure 8.** NR4A nuclear receptors regulate the pro-inflammatory phenotype of PDGFRβ+ cells. (A) Distribution of *Nr4a1, Nr4a2, and Nr4a3* expression, within *tSNE* plot depicted in **Figure 1B**. Transcript counts represent $Log_2$ of gene expression. (B) *Nr4a* mRNA levels in freshly isolated APCs, FIPs, and MLCs, isolated from the gonadal WAT of lean chow-fed male mice. n = 4. (C) *Nr4a* mRNA levels in freshly isolated APCs, FIPs, and MLCs, isolated from the gonadal WAT of male mice following 4 weeks of high-fat diet (HFD) feeding. n = 4. (D) Relative mRNA levels of *Nr4a* family members in cultures of

*Figure 8 continued on next page*

*Figure 8 continued*

FIPs treated with vehicle (PBS) or TNFα (20 ng/ml) for 3 hr. n = 4. * denotes p<0.05 by student's t-test. n = 4. (E) Relative mRNA levels of *Nr4a1* in FIPs 3 days following transduction with retrovirus expressing either *Gfp* or *Nr4a1*. n = 4. (F) Relative mRNA levels of *Nr4a2* in FIPs 3 days following transduction with retrovirus expressing either *Gfp* or *Nr4a2*. n = 4. (G) Relative mRNA levels of *Nr4a3* in FIPs 3 days following transduction with retrovirus expressing either *Gfp* or *Nr4a3*. n = 4. (H) *Cxcl2* expression in FIPs 3 days following transduction with indicated retroviruses and treated with vehicle (PBS) or TNFα (20 ng/ml) for 4 hr. n = 4. (I) *Cxcl10* expression in same cultures shown in (H). (J) *Ccl2* expression in same cultures shown in (H). (K) *Il6* expression in same cultures shown in (H). (L) Relative mRNA levels of *Nr4a1* in FIPs following transduction with retrovirus expressing shRNA targeting *Gfp* (shGFP) (control) or retroviruses individually expressing distinct shRNAs targeting unique regions of *Nr4a1* mRNA (shNr4a1 #1–3). n = 4. (M) *Ccl2* expression in FIPs following transduction with indicated retroviruses and treatment with vehicle (PBS) or TNFα (20 ng/ml) for 3 hr. n = 4. (N) *Tnfa* expression in same cultures shown in (C). (O) *Cxcl2* expression in same cultures shown in (C). (P) *Il6* expression in same cultures shown in (C). * in panels E-P denote p<0.05 by student's t-test in comparison to corresponding treatments of control cells (pMSCV-GFP or shGFP). Bars in all graphs represent mean +SEM.

DOI: https://doi.org/10.7554/eLife.39636.023

FIPs also exert a functional pro-inflammatory phenotype in response to pro-inflammatory stimuli. It is notable that the frequency of FIPs increases following the onset of HFD feeding. This raises an intriguing hypothesis for future studies that perivascular stromal cells can modulate local tissue inflammation. This notion is in line with recent studies indicating that vascular mural cells can serve

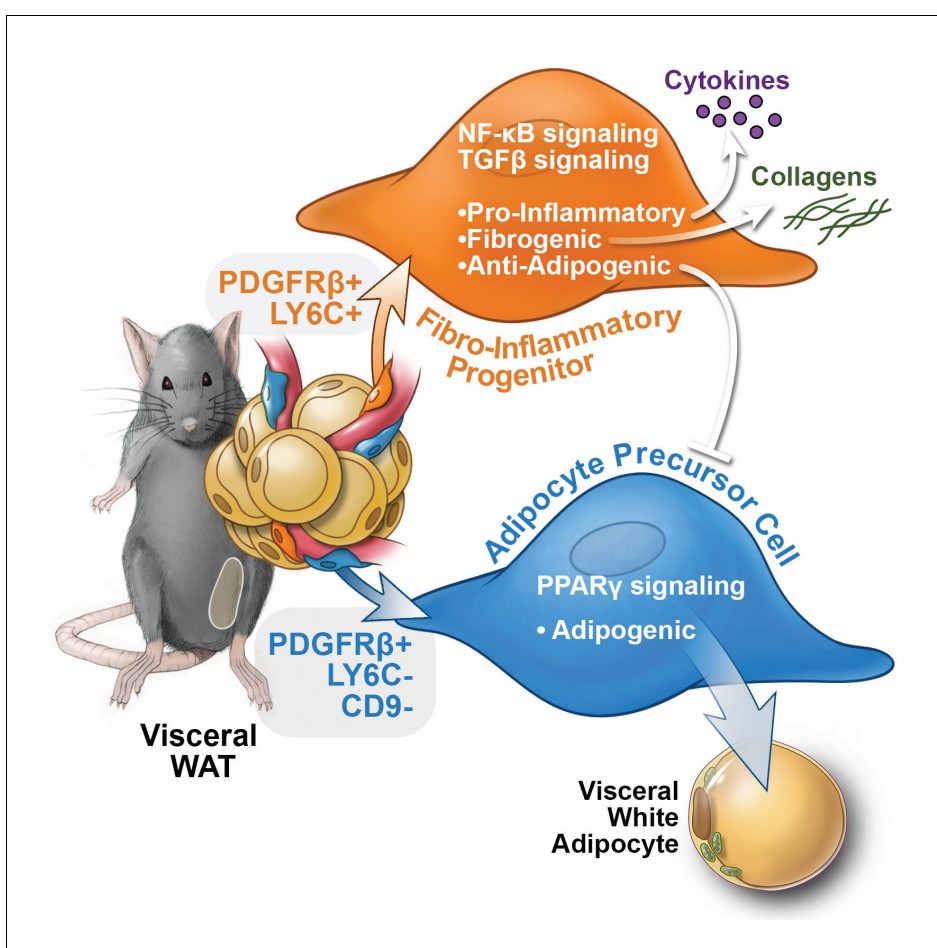

**Figure 9.** Functional heterogeneity of PDGFRβ+ perivascular cells in visceral adipose tissue of mice. The pool of PDGFRβ+ cells in visceral WAT of mice is molecularly and functionally heterogeneous. LY6C- CD9- PDGFRβ+ cells represent visceral adipocyte precursor cells (APCs), whereas LY6C+ PDGFRβ+ cells represent fibro-inflammatory progenitors (FIPs). FIPs are fibrogenic, pro-inflammatory, and inhibit adipocyte differentiation from APCs.
DOI: https://doi.org/10.7554/eLife.39636.024

as 'gatekeepers' of inflammation in the lung (*Hung et al., 2017*). On the other hand, one may expect that the increased frequency of FIPs would also completely blunt the differentiation of APCs in this depot. This is clearly not the case as gonadal WAT in mice is able to expand through adipocyte hyperplasia in the setting of diet-induced obesity. One possibility is that the anti-adipogenic activity of FIPs (rather than the frequency per se) is diminished by local signals in an attempt to facilitate adipogenesis within the depot. Another possibility lies in the spatial distribution of activated FIPs and APCs; their proximity to one another may influence their activity. The identification of these populations from the MuralChaser mice essentially places them within the perivascular compartment of adipose tissue; however, where APCs and FIPs are localized and become activated within the tissue is still unknown. Clearly, additional studies of these cells will be needed in order to determine their precise contribution to WAT inflammation and health in various settings in vivo. Furthermore, it is certainly plausible that FIPs contribute to WAT remodeling beyond fibrosis, inflammation, and adipogenesis.

The ability to isolate functionally distinct subpopulations of mural cells affords the possibility of identifying factors regulating these diverse mural cell phenotypes. Our gene expression analysis revealed the enrichment in mRNA levels of *Nr4a* family members in FIPs. Several studies have implicated NR4A nuclear receptors as modulators of inflammatory signaling; the precise impact of NR4A members on inflammation appears context/cell type specific. In some studies, NR4A family members are observed as being pro-inflammatory (*Pei et al., 2006*). In other settings, NR4A expression is induced in response to pro-inflammatory stimuli but acts as a molecular brake on inflammatory signaling. Our gain- and loss of function studies suggest that FIPs activate *Nr4a* family members in response to pro-inflammatory stimuli to serve as a transcriptional brake on inflammatory cytokine gene expression. This counter-regulatory response may be a mechanism to limit cellular oxidative stress and apoptosis driven by inflammatory signaling (*Rodríguez-Calvo et al., 2017*). Chao et al. previously demonstrated that NR4A family members are potent regulators of adipocyte differentiation (*Chao et al., 2008*). Thus, NR4A members may play multiple roles in controlling of the fate and function of adipose perivascular cells. Additional studies involving the genetic manipulation of *Nr4a* family members in PDGFRβ+ cells in vivo will be needed to elucidate the exact requirements of NR4A members in WAT remodeling in vivo. Importantly, the ability to isolate FIPs and APCs affords the possibility of employing several different types of genomic approaches in an effort to reveal novel molecular mechanisms controlling adipose tissue inflammation, fibrosis, and adipogenesis.

As described above, there has been tremendous interest in elucidating the identity of adipocyte precursors in adult adipose tissue. Pioneering studies from Friedman and colleagues led to a now widely-used strategy to prospectively isolate adipocyte progenitor cells from freshly isolated WAT (*Rodeheffer et al., 2008*). APCs have been isolated on the basis of CD29, SCA-1, and CD34 expression (CD29+ CD34+ SCA-1+ CD31- CD45-). These markers have proven to be quite useful for the selection of APCs from inguinal WAT and other WAT depots; however, a notable observation made here is that SCA-1 expression is in fact enriched in FIPs rather than APCs of the gonadal WAT depot. As such, the selection of cells on the basis of SCA-1 expression from this particular WAT depot yields a functionally heterogeneous population that likely includes FIPs, and perhaps even enriches for these cells. This may explain, at least in part, the notable lack of adipogenic potential that isolated gonadal CD34+ SCA-1+ cells possess in vitro (*Church et al., 2014*). Our prior work pointed to *Zfp423* as a marker of committed preadipocytes; however, our scRNA-seq data reveal that *Zfp423* expression is not confined exclusively to the APC subpopulation of PDGFRβ+ cells; *Pdgfrb*-expressing MLCs and FIPs also express *Zfp423*. Nevertheless, the selection of GFP$^{High}$ PDGFRβ+ cells from gWAT of *Zfp423*$^{GFP}$ reporter mice enriches for committed APCs, perhaps reflecting increased promoter/transgene activity in these cells.

Recent scRNA-seq analyses from Granneman and colleagues provide a cellular atlas of putative adipocyte precursor populations in adipose tissue (*Burl et al., 2018*). Their analyses included all non-hematopoetic, non-endothelial cells of the isolated adipose stromal-vascular fraction. The strength of their approach is that it allows for one to capture both perivascular (PDGFRβ+) and non-perivascular precursor populations. Our approach will identify precursor populations that express *Pdgfrb/rtTA* at the time of the pulse labeling. *Pdgfrb* expression declines as cell undergo differentiation into adipocytes. This means that cells even further committed to the adipocyte lineage (i.e. no longer express *Pdgfrb*) may not be captured through our analysis. Moreover, putative stem cell populations not yet expressing *Pdgfrb* may also be present and not captured (e.g. Pref1$^{rtTA}$ targeted

cells [*Hudak et al., 2014*]). Nevertheless, it is notable that most of the adipocyte precursor populations represented in the study by Burl et al. were indeed captured in our analysis. One cannot rule out the existence of additional adipocyte progenitor populations in any particular adipose depot; however, the congruency of the two independent studies suggests the MuralChaser model can identify and target the major APC populations residing within visceral WAT of mice. Moreover, the selection of LY6C- PDGFRβ+ cells from gWAT using commercially available antibodies represents a refined and convenient strategy to isolate visceral adipocyte precursors from wild type mice or genetic mouse models of interest.

Single-cell transcriptomics has become very useful in revealing molecular heterogeneity amongst seemingly homogenous populations of cells. The challenge, however, is to determine whether molecularly distinct populations of cells represent distinct 'cell types,' or rather 'cell states' which are influenced by their local microenvironment. Here, we reveal molecular heterogeneity amongst PDGFRβ+ cells within visceral white adipose tissue and begin to define the functional differences between the identified subpopulations. Our functional analyses suggest that the properties of visceral FIPs are at least somewhat stable; FIPs are quite limited in adipogenic potential in vitro and upon transplantation. They do not readily activate *Pparg* expression under the conditions examined. The phenotype of visceral APCs may be less stable. Visceral APCs have the potential to adopt characteristics of FIPs. In vitro and in vivo following HFD, APCs can activate the expression of pro-inflammatory cytokines. A caveat to most of our functional studies is that the cells are studied outside their native microenvironment. Under some physiological conditions, it is certainly plausible that multiple PDGFRβ+ subpopulations give rise in vivo to adipocytes; such adipocytes might even possess unique functional characteristics. Our prior lineage-tracing studies using the MuralChaser model clearly established that adipocytes emerge from *Pdgfrb*-expressing cells; efforts to define the relative contribution of individual mural cell subpopulations will require more precise lineage-tracing approaches with more specific Cre drivers. Moreover, our current studies cannot exclude the possibility that even further heterogeneity exists amongst PDGFRβ+ cells or within the identified subpopulations. Deeper sequencing, refined analyses, and further functional studies may unveil even more heterogeneity than appreciated.

It is noteworthy that *Pdgfrb* is expressed in a subset of WAT associated mesothelial cells and that *Pdgfrb*-expressing mesothelial cells express *Pparg* and some level of *Zfp423*. As described above, several lines of evidence point to a lineage relationship between embryonic mesothelial cells, APCs, and perivascular stromal cells within the visceral compartment (*Chau et al., 2014*; *Rinkevich et al., 2012*). PDGFRβ+ MLCs did not exhibit robust adipogenic potential under the culture conditions utilized here, despite their expression of *Pparg* isoform two and *Zfp423*. Additional signals may be needed in order to drive adipocyte differentiation from these cells. Alternatively, this subpopulation of mesothelial cells may represent developmental intermediates between mesothelial cells and perivascular progenitors. Further insight into the functional significance of these various stromal subpopulations, their developmental origins, and their cellular plasticity will require additional genetic tools to manipulate individual populations selectively in vivo. Nevertheless, the molecular profiles obtained for FIPs and APCs from visceral WAT, along with the strategies to isolate these cells, will facilitate the study of physiological visceral WAT remodeling in vivo. Ultimately, unraveling the cellular and molecular determinants of WAT expansion and remodeling may lead to strategies to improve adipose tissue function and defend against metabolic disease.

## Materials and methods

**Key resources table**

| Reagent type (species) or resource | Designation | Source or reference | Identifiers | Additional information |
|---|---|---|---|---|
| Antibody | anti-guinea pig Alexa 647 | Invitrogen | RRID:AB_141882 | 1:200 |
| Antibody | anti-chicken Alexa 488 | Invitrogen | RRID:AB_142924 | 1:200 |
| Antibody | anti-GFP | Abcam | RRID:AB_300798 | 1:700 |
| Antibody | anti-Perilipin | Fitzgerald | RRID:AB_1288416 | 1:1500 |

*Continued on next page*

*Continued*

| Reagent type (species) or resource | Designation | Source or reference | Identifiers | Additional information |
|---|---|---|---|---|
| Antibody | CD24-APC | eBioscience | RRID:AB_10852841 | 1:400 |
| Antibody | CD31-FITC | Biolegend | RRID:AB_312900 | 1:400 |
| Antibody | CD31-PerCP/Cy5.5 | Biolegend | RRID:AB_10612742 | 1:400 |
| Antibody | CD34-APC | Biolegend | RRID:AB_10553895 | 1:400 |
| Antibody | CD38-FITC | Biolegend | RRID:AB_312926 | 1:400 |
| Antibody | CD45-FITC | Biolegend | RRID:AB_312973 | 1:400 |
| Antibody | CD45-PerCP/Cy5.5 | Biolegend | RRID:AB_893344 | 1:400 |
| Antibody | CD9-APC | eBioscience | RRID:AB_10669565 | 1:400 |
| Antibody | CD9-FITC | Biolegend | RRID:AB_1279321 | 1:400 |
| Antibody | FC Block | BD Biosciences | RRID:AB_394657 | 1:200 |
| Antibody | LY6C-APC | Biolegend | RRID:AB_1732076 | 1:400 |
| Antibody | LY6C-BV421 | Biolegend | RRID:AB_2562178 | 1:400 |
| Antibody | PDGFRα-APC | Biolegend | RRID:AB_2043970 | 1:200 |
| Antibody | PDGFRβ-APC | Biolegend | RRID:AB_2268091 | 1:50 |
| Antibody | PDGFRβ-PE | Biolegend | RRID:AB_1953271 | 1:50 |
| Antibody | SCA-1-APC | Biolegend | RRID:AB_313348 | 1:400 |
| Chemical compound, drug | Trypsin | Corning | 25–052 Cl | |
| Chemical compound, drug | BrdU | Sigma | B5002 | |
| Chemical compound, drug | BSA | Fisher Scientific | BP1605 | |
| Chemical compound, drug | Collagenase D | Roche | 11088882001 | |
| Chemical compound, drug | Dexamethosone | Sigma | D4902 | |
| Chemical compound, drug | DMEM with 1 g/L glucose, L-glutamine, and sodium pyruvate | Corning | 10–014-CV | |
| Chemical compound, drug | DMEM/F12 with GlutaMAX | Gibco | 10565–018 | |
| Chemical compound, drug | FBS | Sigma | 12303C | |
| Chemical compound, drug | FGF basic | R and D Systems | 3139-FB-025/CF | |
| Chemical compound, drug | Gentamicin Reagent (50 mg/ml) | Gibco | 15750–060 | |
| Chemical compound, drug | Harris Eosin Solution | Sigma | HT110116 | |
| Chemical compound, drug | Harris Hematoxylin Solution | Sigma | HHS16 | |
| Chemical compound, drug | HBSS | Sigma | H8264 | |
| Chemical compound, drug | Insulin | Sigma | I6634 | |
| Chemical compound, drug | Isobutylmethyxanthine | Sigma | I7018 | |
| Chemical compound, drug | ITS Premix | BD Bioscience | 354352 | |
| Chemical compound, drug | L-ascorbic acid-2-2phosphate | Sigma | A8960 | |
| Chemical compound, drug | Lipofectamine LTX | Invitrogen | 15338100 | |
| Chemical compound, drug | Lipopolysaccharides from Escherichia coli O111:B4 | Sigma | L3024 | |
| Chemical compound, drug | Matrigel Growth Factor Reduced Membrane Matrix | Corning | 354230 | |
| Chemical compound, drug | MCDB201 | Sigma | M6770 | |
| Chemical compound, drug | M-MLV RT | Invitrogen | 28025013 | |
| Chemical compound, drug | Oil Red O | Sigma | O0625 | |
| Chemical compound, drug | PBS | Sigma | D8537 | |
| Chemical compound, drug | pCMV-VSV-G | Addgene | 8454 | |

*Continued*

| Reagent type (species) or resource | Designation | Source or reference | Identifiers | Additional information |
|---|---|---|---|---|
| Chemical compound, drug | Penicillin Streptomycin Solution | Corning | 30–001 Cl | |
| Chemical compound, drug | Polybrene | Sigma | TR-1003 | |
| Chemical compound, drug | psPAX2 | Addgene | 12260 | |
| Chemical compound, drug | Random Primers | Invitrogen | 48190011 | |
| Chemical compound, drug | Recombinant Human TGFβ−1 | R and D Systems | 240-B-002 | |
| Chemical compound, drug | Recombinant Murine TNFα | PeproTech | 315-01A | |
| Chemical compound, drug | Red Blood Cell Lysing Buffer Hybri-Max | Sigma | R7757 | |
| Chemical compound, drug | SYBR Green PCR Master Mix | Applied Biosystems | 4309155 | |
| Chemical compound, drug | Trizol | Invitrogen | 15596018 | |
| Commercial assay or kit | Chromium i7 Multiplex Kit, 96 rxns | 10X Genomics | 120262 | |
| Commercial assay or kit | Chromium Single Cell 3' Library and Gel Bead Kit v2, 16 rxns | 10X Genomics | 120237 | |
| Commercial assay or kit | Chromium Single Cell A Chip Kit, 48 rxns | 10X Genomics | 120236 | |
| Commercial assay or kit | Dynabeads MyOne Silane | Thermo Fisher Scientific | 37002D | |
| Commercial assay or kit | FITC BrdU Flow Kit | BD Biosciences | 559619 | |
| Commercial assay or kit | RNAqueous-Micro Total RNA Isolation Kit | Invitrogen | AM1931 | |
| Commercial assay or kit | SPRIselect | Beckman Coulter | B23317 | |
| Other | 100 µm cell strainer | Falcon | 352360 | |
| Other | 12-well plate | Corning | 356500 | |
| Other | 40 µm cell strainer | Falcon | 352340 | |
| Other | 48-well plate | Falcon | 353230 | |
| Other | doxycyline-containing chow diet (600 mg/kg doxycycline) | Bio-Serv | S4107 | |
| Other | high-fat diet (60% kcal% fat) | Research Diets | D12492i | |
| Software, algorithm | Cell Ranger v2.1.0 | 10X Genomics | NA | |
| Software, algorithm | FlowJo V10 | FlowJo | RRID:SCR_008520 | |
| Software, algorithm | Gene Set Enrichment Analysis v3.0 | Broad Institute | RRID:SCR_003199 | |
| Software, algorithm | Graphpad Prism 7 | Graphpad | RRID:SCR_002798 | |
| Software, algorithm | R Studio v3.3.2 | RStudio | RRID:SCR_000432 | |
| Software, algorithm | Readr v1.1.0 | NA | NA | |
| Software, algorithm | Seurat v2.1.0 | Satija Lab | RRID:SCR_016341 | |
| Strain, strain background (M. musculus, C57BL/6) | C57BL/6 | Charles River Laboratories | RRID:IMSR_CRL:27 | |
| Strain, strain background (M. musculus, C57BL/6) | *Pdgfrb*rtTA | Jackson Laboratories | RRID:IMSR_JAX:028570 | |
| Strain, strain background (M. musculus, C57BL/6) | *Rosa26R*mT/mG | Jackson Laboratories | RRID:IMSR_JAX:007676 | |
| Strain, strain background (M. musculus, C57BL/6) | *TRE-Cre* | Jackson Laboratories | RRID:IMSR_JAX:006234 | |
| Strain, strain background (M. musculus, C57BL/6) | *Zfp423*GFP | Other | *Zfp423*GFPB6 | PMID: 26626462 |

## Animals and diets

All animal experimens were performed according to procedures approved by the UTSW Animal Care and Use Committee. MuralChaser mice were derived from breeding $Pdgfrb^{rtTA}$ (JAX 028570), $TRE\text{-}Cre$ (JAX 006234), and $Rosa26R^{mT/mG}$ (JAX 007676) mice, as previously described (*Vishvanath et al., 2016*). Wildtype C57BL/6 mice mice were purchased from Charles River Laboratories and $Zfp423^{GFPB6}$ mice were described previously (*Vishvanath et al., 2016*). Mice were maintained on a 12 hr light/dark cycle in a temperature-controlled environment (22°C) and given free access to food and water. Mice were fed a standard rodent chow diet, doxycyline-containing chow diet (600 mg/kg doxycycline, Bio-Serv, S4107), or high-fat diet (60% kcal% fat, Research Diets, D12492i). For all experiments involving MuralChaser mice, 6 weeks-old mice were fed doxycycline-containing chow for 9 days, and then standard chow for additional 5 days before analysis (doxcycy-line washout). For the high fat diet feeding experiments, mice were placed on the high fat diet beginning at 8 weeks of age.

## Histological analysis

Adipose tissues were harvested from perfused (4% paraformaldehyde) mice. Paraffin embedding and tissue sectioning was performed by the Molecular Pathology Core Facility at UTSW. Indirect immunofluorescence was performed as previously described (*Vishvanath et al., 2016*). Antibodies used for immunofluorescence include: anti-GFP 1:700 (Abcam ab13970), anti-perilipin 1:1500 (Fitzgerald 20R-PP004), anti-chicken Alexa 488 1:200 (Invitrogen), and anti-guinea pig Alexa 647 1:200 (Invitrogen). Hematoxylin and eosin staining was performed according to manufacturer's instructions.

## Gene expression analysis by quantitative PCR

RNA was isolated from freshly sorted cells using RNAqueous-Micro Total RNA Isolation Kit (Thermo Fisher Scientific) or from cell cultures using Trizol, according to manufacturer's instructions. cDNA was synthesized using M-MLV Reverse Transcriptase (Invitrogen) and Random Primers (Invitrogen). Relative mRNA levels were determined by quantitative PCR using SYBR Green PCR Master Mix (Applied Biosystems). Values were normalized to Rps18 levels using the $\Delta\Delta$-Ct method. Unpaired Student's t-test was used to evaluate statistical significance. All primer sequences are listed within *Table 4*.

## Isolation of adipose stromal vascular fraction (SVF) and flow cytometry

Adipose tissue was minced with scissors in a 1.5 mL tube containing 200 µL of digestion buffer (1X HBSS, 1.5% BSA, and 1 mg/mL collagenase D) and then transferred to a 50 mL Falcon tube containing 10 mL digestion buffer. The mixture was incubated in a 37°C shaking water bath for 1 hr. The solution of digested tissue was passed through a 100 µm cell strainer, diluted to 30 mL with 2% FBS in PBS, and centrifuged at 500 x g for 5 min. The supernatant was aspirated and red blood cells in the SVF pellet were lysed by brief incubation in 1 mL RBC lysis buffer (Sigma). Next, the mixture was diluted to 10 mL with 2% FBS in PBS, passed through a 40 µm cell strainer, and then centrifuged at 500 x g for 5 min. The supernatant was aspirated, and cells were resuspended in blocking buffer (2% FBS/PBS containing anti-mouse CD16/CD32 Fc Block (1:200)). Primary antibodies were added to the cells in blocking buffer for 15 min at 4°C in the dark. After incubation, the cells were washed once with 2% FBS/PBS and then resuspended in 2% FBS/PBS for sorting. Cells were sorted for collection using a BD Biosciences FACSAria cytometer or analyzed using a BD Biosciences LSR II cytometer (UTSW Flow Cytometry Core Facility). Flow cytometry plots were generated with FlowJo (V10).

## BrdU assays

Eight week-old mice were administered 0.8 mg/mL BrdU in drinking water (replaced fresh every 2 days) and placed on chow or high fat diet for 1 week. At the end of the treatment period, adipose tissue SVF was isolated as described above and stained with the following antibodies: CD31, CD45, PDGFRβ, LY6C, and CD9. Anti-BrdU staining of fixed cells was then conducted using the BrdU Flow Kit (BD Biosciences 559619), according to the manufacturer's protocol.

**Table 4.** Sequences of qPCR primers used in this study.

| Gene | Forward 5'—3' | Reverse 5'—3' |
| --- | --- | --- |
| Abcg1 | CAGCCTCTGGAGGGATTCTTT | ATCCCACGGCACTCTCACTTA |
| Adipoq | AGATGGCACTCCTGGAGAGAA | TTCTCCAGGCTCTCCTTTCCT |
| Adipsin | CTACATGGCTTCCGTGCAAGT | AGTCGTCATCCGTCACTCCAT |
| Agpat2 | CGAAGCTCTTCACCTCAGGAA | TCTGTAGAAAGGTGGCCCTCA |
| Agt | GTTCTGGGCAAAACTCAGTGC | GAGGCTCTGCTGCTCATCATT |
| Car3 | CTTTGGAGAGGCTCTGAAGCA | ATCTGGAACTCGCCTTTCTCC |
| Ccl2 | CCACAACCACCTCAAGCACTTC | AAGGCATCACAGTCCGAGTCAC |
| Cd24 | CCTCCTCCTGTGGCTTTAGGTCTG | GGTGCTTGTGGTGAGTGAGAAACG |
| Cd34 | TGTGAAAAGGAGGAGGCTGAG | GTTTGCTGGGAAGTTCTGTGC |
| Cd36 | GAGTTGGCGAGAAAACCAGTG | GAGAATGCCTCCAAACACAGC |
| Cd38 | GCACCTTTGGAAGTGTGGAAG | CATGCGTTACTGGAAGCTCCT |
| Cebpa | CAAGAACAGCAACGAGTACCG | GTCACTGGTCAACTCCAGCAC |
| Chst4 | CAGCAAACAGCATCTGTGGAG | CTTCGGAAAGATGTGGACAGG |
| Col1a1 | AGATGATGGGGAAGCTGGCAA | AAGCCTCGGTGTCCCTTCATT |
| Col2a1 | AGAACCTGGTACCCCTGGAAA | ACCACCAGCCTTCTCGTCATA |
| Col3a1 | ATTCTGCCACCCCGAACTCAA | ACAGTCATGGGGCTGGCATTT |
| Col5a1 | TGTCATGTTTGGCTCCCGGAT | AGTCATAGGCAGCTCGGTTGT |
| Cxcl10 | CTCAGGCTCGTCAGTTCTAAGT | CCCTTGGGAAGATGGTGGTTAA |
| Cxcl14 | TGGACGGGTCCAAGTGTAAGT | TCCTCGCAGTGTGGGTACTTT |
| Cxcl2 | ACTAGCTACATCCCACCCACAC | GCACACTCCTTCCATGAAAGCC |
| Dact2 | AGCCCCCTAAAGGAAGAAACC | GGTCCTTGGCCACAGTCATTA |
| Efhd1 | GGCCGCTCTAAGGTCTTCAAT | GTCAATAAAGCCGTCCCTTCC |
| F3 | AAGGATGTGACCTGGGCCTAT | AGTTGGTCTCCGTCTCCATGA |
| Fabp5 | GATGGGAAGATGATCGTGGAG | AACTCCTGTCCAGGATGACGA |
| Fn1 | GAGAGCACACCCGTTTTCATC | GGGTCCACATGATGGTGACTT |
| Glut4 | ATCTTGATGACCGTGGCTCTG | GCTGAAGAGCTCTGCCACAAT |
| Hspd1 | GCACGATCTATTGCCAAGGAG | TCTTCAGGGGTTGTCACAGGT |
| Il1b | GCAACTGTTCCTGAACTCAACT | ATCTTTTGGGGTCCGTCAACT |
| Il6 | AAGCCAGAGTCCTTCAGAGAGA | ACTCCTTCTGTGACTCCAGCTT |
| Krt18 | GCTGCAGCTGGAGACAGAAAT | GTCAATCCAGAGCTGGCAATC |
| Krt8 | GAATGGCCACTGAAGTCCTTG | AGTTCCCTGCACTCTGCCATA |
| Lox | TCGCTACACAGGACATCATGC | ATGTCCAAACACCAGGTACGG |
| Loxl2 | ACCCACGTCTGTATTCCATGC | CATCCAAGTCTTCAGCCATCC |
| Lrrn1 | CAACATGGGAGAGCTGGTTTC | GCACACTACGGAAAGCCAAAC |
| Ly6a | ACACAGCCAGCACAGTGAAGA | CAGGGGGACATTCAGGATACA |
| Ly6c1 | ACTGTGCCTGCAACCTTGTCT | GGCCACAAGAAGAATGAGCAC |
| Mmd2 | ATCTGGGAGCTGATGACAGGA | AGTGGGTACCAGCACCAAATG |
| Nos2 | CCTCTGGTCTTGCAAGCTGAT | ACTCGTACTTGGGATGCTCCA |
| Nov | GTTCCAAGAGCTGTGGAATGG | CTCTTGTTCACAAGGCCGAAC |
| Nr4a1 | TCTCTGGTTCCCTGGACGTTA | ACCGGGTTTAGATCGGTATGC |
| Nr4a1-1317 | TGCCTCCCCTACCAATCTTCT | TAACGTCCAGGGAACCAGAGA |
| Nr4a1-1468 | TCTCTGGTTCCCTGGACGTTA | ACCGGGTTTAGATCGGTATGC |
| Nr4a1-1877 | CGCATTGCTAGCTGTCTGAAAG | AATAGGTGGAGGGGGTACCA |
| Nr4a2 | ACACAGCGGGTCGGTTTACTA | ATGCGTAGTGGCCACGTAGTT |

*Table 4 continued on next page*

*Table 4 continued*

| Gene | Forward 5'—3' | Reverse 5'—3' |
| --- | --- | --- |
| Nr4a3 | ACTTGCAGAGCCTGAACCTTG | TTGGTGCATAGCTCCTCCACT |
| Pde11a | CGAGCTTGTCAGGAAAGGAGA | TTCAGCCACCTGTCTGGAGAT |
| Pdgfra | ATCAGCTTGGCTCTTCCCTTC | TATAGCTTCCTGCTCCCGTCA |
| Pdgfrb | AGGGGGTGATAGCTCACATCA | AGCCATAACACGGACAGCAAC |
| Pkhd11b | CAGATTGGGACAGAAGCATCC | ACAGGAATAGGCAGACCGTGA |
| Plin1 | CAGTTCACAGCTGCCAATGAG | ATGGTGCCCTTCAGTTCAGAG |
| Pparg isoform 1 | TGAAAGAAGCGGTGAACCACT | TGGCATCTCTGTGTCAACCAT |
| Pparg isoform 2 | GCATGGTGCCTTCGCTGA | TGGCATCTCTGTGTCAACCATG |
| Rbp4 | TCTGTGGACGAGAAGGGTCAT | TGTCTGCACACACTTCCCAGT |
| Rps18 | CATGCAAACCCACGACAGTA | CCTCACGCAGCTTGTTGTCTA |
| Stmn4 | ACCTGAACTGGTGCGTCATCT | CTTGGGAGGGAGGCATTAAAC |
| Tgfb1 | TTTAGGAAGGACCTGGGTTGG | TGTTGGTTGTAGAGGGCAAGG |
| Tgfb2 | GGTGTTGTTCCACAGGGGTTA | CGGTCCTTCAGATCCTCCTTT |
| Tnfa | GAAAGGGGATTATGGCTCAGG | TCACTGTCCCAGCATCTTGTG |
| Upk3b | GCTTGGCCAACTTAACCTCCT | TGCTGCGTTCTCTGAAGTCTG |
| Zfp423 | CAGGCCCACAAGAAGAACAAG | GTATCCTCGCAGTAGTCGCACA |

DOI: https://doi.org/10.7554/eLife.39636.025

## Single-cell RNA-sequencing and analysis

Six-week-old male MuralChaser mice were fed doxycycline-containing chow diet for 9 days, followed by standard chow diet for 5 days. Following the 5 day washout period, gonadal WAT was isolated and digested as described above. tdTomato- mGFP+ cells were collected by FACS. Single cell library preparation was performed using the 10X Genomics Single Cell 3' v2 according to the manufacturer's instructions.

After FACS isolation of gonadal WAT tdTomato- mGFP+ cells from MuralChaser mice, 10,000 cells were partitioned into droplets containing a barcoded bead, a single cell, and reverse transcription enzyme mix using the GemCode instrument. This was followed by cDNA amplification, fragmentation, end repair and A-tailing, adaptor ligation, and index PCR. Cleanup and size selection were performed using Dynabeads MyOne Silane beads (Thermo Fisher Scientific) and SPRIselect Reagent beads (Beckman Coulter). Libraries were sequenced on an Illumina NextSeq 500 High Output (400M) by the UT Southwestern McDermott Center Next Generation Sequencing Core. 75 paired-end reads were obtained using one flow cell with the following length input: 26 bp Read 1, 66 bp Read 2, 0 bp Index 1, and 0 bp Index 2.

Cell Ranger software (v2.1.0) was used to perform demultiplexing, aligning reads, filtering, clustering, and gene expression analyses, using default parameters. We recovered 1378 cells with a median UMI count of 10,879 per cell, a mean reads per cell of 277,212, and a median genes per cell of 3278. In order to ensure that our analysis was restricted to genetically marked *Pdgfrb*-expressing cells, we filtered the cells based on expression of *tdTomato* ($\leq$0) and *GFP* (>0) to only include cells in the final analysis that were devoid of *tdTomato* mRNA and expressed *GFP* transcript. After this screening, we obtained a total of 1,045 cells for the analysis shown in *Figure 1*. The Cell Ranger data was imported into Loupe Cell Browser Software (v1.0.5) for t-distributed stochastic neighbor embedding (*tSNE*) based clustering, heatmap generation, and gene expression distribution plots. The Cell Ranger files were imported into R Studio (v3.3.2) and the Seurat (v2.1.0) and Readr (v1.1.0) packages were used to generate gene cluster text (GCT) and categorical class (CLS) files, using the clustering generated from Cell Ranger (*k*-means = 4 for the analysis in *Figure 1* and *k*-means = 3 for the analysis in *Figure 2*). The GCT and CLS files were input into Gene Set Enrichment Analysis (GSEA) (v3.0) using the Java GSEA implementation with default parameters. The single cell RNA-sequencing experiment was repeated using cells isolated from pooled gonadal WAT from five additional MuralChaser mice to validate the identification of APCs, FIPs, and MLCs (*Figure 1—figure*

*supplement 1*). The raw sequencing data from *Figures 1* and *2* has been deposited to Gene Expression Omnibus (https://www.ncbi.nlm.nih.gov/geo/query/acc.cgi?acc=GSE111588).

## In vitro adipocyte differentiation

For all cellular assays, 4–6 weeks-old C57BL/6 mice were utilized. PDGFRβ+ subpopulations were isolated from gonadal WAT SVF as described above. For spontaneous adipogenesis assays, sorted cells were plated in 48-well plates at a density of $4 \times 10^4$ cells/well in growth media containing 2% FBS and ITS supplement [60% pH7–7.4 low glucose DMEM, 40% pH 7.25 MCDB201 (Sigma M6770), 1% ITS premix (Insulin-Transferrin-Selenium) (BD Bioscience 354352), 0.1 mM L-ascorbic acid-2-2phosphate (Sigma A8960-5G), 10 ng/mL FGF basic (R and D Systems 3139-FB-025/CF), Pen/Strep, and gentamicin] and incubated at 37°C in 10% $CO_2$. Media was replaced every other day. For induced adipogenesis, confluent cultures of FIPs were treated with an adipogenesis induction cocktail (growth media supplemented with 5 mg/ml insulin, 1 μM dexamethasone, 0.5 mM isobutylmethyxanthine, ±1 μM rosiglitazone) for 48 hr. After 48 hr., the cells were maintained in growth media.

## Oil red O staining

Cells were fixed in 4% PFA for 15 min at room temperature and washed twice with water. Cells were incubated in Oil Red O working solution (2 g Oil red O in 60% isopropanol) for 10 min to stain accumulated lipids. Cells were then washed three times with water and bright field images were acquired using a document scanner or with a Keyence BZ-X710 Microscope.

## Mesothelial cell isolation

Adipose associated mesothelial cells were isolated as described previously (*Darimont et al., 2008*). Epididymal adipose depots were harvested from 6 weeks-old male MuralChaser mice treated with doxycyline as described above. Intact whole adipose depots were placed in 10 mL of PBS containing 0.25% trypsin for 20 min at 37°C with continuous end over end rotation. Next adipose tissues were removed, and remaining solution containing isolated cells was centrifuged at 600 x g for 5 min. The media was aspirated, and the cell pellet was resuspended in growth media (10% FBS in DMEM/F12 (Invitrogen)) and plated in a 12-well collagen-coated plate. The cells were incubated at 37°C in 10 $CO_2$ and the media was replaced daily. Images were obtained using a Leica DMIL LED microscope and a Leica DFC3000g camera.

## Cellular assays

To assess in vitro proliferation, sorted cells (APCs and FIPs) were plated at a density of $5 \times 10^3$ cells/well in a 48-well plate containing 2% FBS in ITS Media. Cell numbers were assessed in parallel wells every 2 days by cell counting with a hemocytometer. To study the impact of conditioned media on adipogenesis, media from equally confluent cultures of APCs, FIPs, and MLCs was harvested and placed onto APCs beginning 48 hr after culture in a 1:1 ratio with 2% FBS in ITS media. Cells were harvested at the indicated time points for RNA expression analysis. Images were obtained using a Leica DMIL LED microscope and a Leica DFC3000g camera.

Bone marrow derived macrophages (BMDMs) were derived from bone-marrow stem cells (BMSCs) isolated from the femurs and tibias of male mice, as previously described (*Shan et al., 2017*). BMSCs were maintained in differentiation medium derived from L929 cells for 7 days to allow for macrophage differentiation. In parallel, adipose tissue SVF was isolated and PDGFRβ+ subpopulations were sorted as described above. Sorted cells (APCs, FIPs, and MLCs) were plated in a 48-well plate at $4 \times 10^4$ cells/well in 2% FBS in ITS media. 48 hr later, the adipose-derived cells were treated with vehicle (PBS) or LPS (100 ng/ml) for 3 hr. Next the adipose-derived cells were washed with PBS and fresh media was added. 24 hr later, the conditioned media was harvested and placed on the BMDMs (at day 7) in a 1:1 ratio with 2% FBS in ITS media. After a 3 hr incubation, the BMDMs were harvested for RNA analysis.

For TGFβ treatments, sorted cells were plated in 48-well plates at $2 \times 10^4$ cells/well in 2% FBS in ITS media. 24 hr later, vehicle (PBS) or 1 ng/mL recombinant TGFβ was added to the media for 3 days prior to harvest. The media was replaced daily under this period. For the LPS and TNFα treatments, cells were plated in 48-well plates at $4 \times 10^4$ cells/well in 2% FBS in ITS media. 48 hr later,

the cells were treated with vehicle (PBS), LPS (100 ng/ml), or TNFα (20 ng/ml). After 3 hr of treatment, cells were harvested for RNA isolation.

## Cell transplantation assays

80,000 cells (APCs and FIPs) collected by FACS were suspended in 100 µL transplantation media (50% Matrigel in PBS, supplemented with 2 ng/mL FGF) and injected subcutaneously into the remnant inguinal WAT region of 3 month old lipodystrophic mice (*Adiponectin*-Cre; *Pparg*$^{loxP/loxP}$). Three weeks later, the remnant inguinal WAT depots were harvested for histology.

## Retroviral production and transduction of primary cells

The pMSCV-Nr4a1, pMSCV-Nr4a2, and pMSCV-Nr4a3 plasmids were previously reported (kind gift from Dr. P. Tontonoz) (*Chao et al., 2008*). Retroviral production and packaging in phoenix cells was performed as previously described (*Shao et al., 2016*). Briefly, 10 µg of the pMSCV overexpression plasmids were co-transfected with 5 µg gag-pol and 5 µg VSV-g plasmids into phoenix packaging cells using Lipofectamine LTX (Thermo Fisher Scientific), according to the manufacturer's protocol. APCs and FIPs were transduced with diluted virus (1:10 ratio) in 2% FBS/ITS media containing 8 µg/ml polybrene (Sigma). Following 16 hr of incubation with indicated viruses, cells were returned to 2% FBS/ITS media and assayed for TNFα responsiveness as indicated.

Double-stranded DNA sequence encoding the shRNA targeting *Nr4a1* was selected from the Broad Institute public database (https://portals.broadinstitute.org/gpp/public/) and cloned into the AgeI/EcoRI sites of the pMKO-1 U6 retroviral vector. The pMKO-1 vector expressing shRNA targeting GFP was used as a negative control. The selected DNA sequences encoding the *Nr4a1* shRNAs used in the study are as follows: shNr4a1-1317 forward oligonucleotide,
5′-CCGGTGCCGGTGACGTGCAACAATTCTCGAGAATTGTTGCACGTCACCGGCATTTTTG-3′;
shNr4a1-1317 reverse oligonucleotide,
5′-AATTCAAAAATGCCGGTGACGTGCAACAATTCTCGAGAATTGTTGCACGTCACCGGCA-3′.
shNr4a1-1468 forward oligonucleotide,
5′-CCGGCGCCTGGCATACCGATCTAAACTCGAGTTTAGATCGGTATGCCAGGCGTTTTTG-3′;
shNr4a1-1468 reverse oligonucleotide,
5′-AATTCAAAAACGCCTGGCATACCGATCTAAACTCGAGTTTAGATCGGTATGCCAGGCG-3′.
shNr4a1-1877 forward oligonucleotide,
5′-CCGGCTATTGTGGACAAGATCTTTACTCGAGTAAAGATCTTGTCCACAATAGTTTTTG-3′;
shNr4a1-1877 reverse oligonucleotide,
5′-AATTCAAAAACTATTGTGGACAAGATCTTTACTCGAGTAAAGATCTTGTCCACAATAG-3′.

## Statistical analysis

All data were expressed as the mean +SEM. We used GraphPad Prism 7.0 (GraphPad Software, Inc., La Jolla, CA, USA) to perform the statistical analyses. For comparisons between two independent groups, a Student's *t*-test was used and $p < 0.05$ was considered statistically significant. For in vitro studies, we estimated the approximate effect size based on independent preliminary studies. Studies designed to characterize an in vitro difference in gene expression were estimated to have a slightly larger effect size of 30% with assumed 15% standard deviation of group means. To detect this difference at a power of 80% and an alpha of 0.05, we predicted we would need four independent replicates per group. We estimated this effect size based on independent preliminary studies. Statistical information, including p values, samples sizes, and repetitions, for all datasets are provided in *Supplementary file 1*.

## Acknowledgements

The authors are grateful to P Scherer for critical reading of the manuscript and members of the UTSW Touchstone Diabetes Center for useful discussions. The authors thank the UTSW Animal Resource Center, Metabolic Phenotyping Core, Pathology Core, Live Cell Imaging Core, Bioinformatics Core Facility, Flow Cytometry Core, and McDermott Sequencing Center, for excellent guidance and assistance with experiments performed here. This study was supported by the NIH NIGMS training grant T32 GM008203 and F31DK113696 to CH, NIDDK R01 DK104789 to RKG, the American Heart Association postdoctoral fellowship 16POST26420136 to MS, NIH NIGMS training grant

T32 GM008203 to ALG, NIDDK DK098277 and DK110497 to DWS, and CPRIT RR140023, NIGMS DP2GM128203, and Welch Foundation I-1926–20170325 to GCH

## Additional information

### Funding

| Funder | Grant reference number | Author |
|---|---|---|
| National Institutes of Health | T32 GM008203 | Chelsea Hepler |
| National Institutes of Health | F31DK113696 | Chelsea Hepler |
| American Heart Association | 16POST26420136 | Mengle Shao |
| National Institutes of Health | T32 GM008203 | Alexandra L Ghaben |
| National Institutes of Health | DK098277 | Douglas Strand |
| National Institutes of Health | DK110497 | Douglas Strand |
| Cancer Prevention and Research Institute of Texas | RR140023 | Gary Hon |
| National Institute of General Medical Sciences | DP2GM128203 | Gary Hon |
| Welch Foundation | I-1926-20170325 | Gary Hon |
| National Institutes of Health | DK104789 | Rana K Gupta |

The funders had no role in study design, data collection and interpretation, or the decision to submit the work for publication.

### Author contributions

Chelsea Hepler, Conceptualization, Data curation, Formal analysis, Supervision, Funding acquisition, Investigation, Visualization, Methodology, Writing—original draft, Project administration, Writing—review and editing; Bo Shan, Conceptualization, Data curation, Formal analysis, Investigation, Methodology, Writing—review and editing; Qianbin Zhang, Mengle Shao, Formal analysis, Investigation, Writing—review and editing; Gervaise H Henry, Lavanya Vishvanath, Resources, Formal analysis, Investigation, Methodology, Writing—review and editing; Alexandra L Ghaben, Resources, Investigation, Methodology, Writing—review and editing; Angela B Mobley, Resources, Formal analysis, Supervision, Investigation, Methodology; Douglas Strand, Resources, Formal analysis, Supervision, Methodology, Writing—review and editing; Gary C Hon, Resources, Formal analysis, Supervision, Investigation, Methodology, Writing—review and editing; Rana K Gupta, Conceptualization, Formal analysis, Supervision, Funding acquisition, Investigation, Writing—original draft, Project administration, Writing—review and editing

### Author ORCIDs

Chelsea Hepler http://orcid.org/0000-0001-9938-382X
Gervaise H Henry http://orcid.org/0000-0001-7772-9578
Mengle Shao http://orcid.org/0000-0002-5488-9904
Alexandra L Ghaben http://orcid.org/0000-0001-7377-7820
Douglas Strand http://orcid.org/0000-0002-0746-927X
Rana K Gupta http://orcid.org/0000-0002-9001-4531

### Ethics

Animal experimentation: This study was performed in strict accordance with the recommendations in the Guide for the Care and Use of Laboratory Animals of the National Institutes of Health. All of the animals were handled according to approved institutional animal care and use committee (IACUC) protocols (#2018-102384 and #2012-0072) of UT Southwestern Medical Center.

Decision letter and Author response
Decision letter https://doi.org/10.7554/eLife.39636.031
Author response https://doi.org/10.7554/eLife.39636.032

## Additional files

### Supplementary files

• Supplementary file 1. Table of statistical data (exact *p* values and sample/cohort sizes for each dataset in the study).
DOI: https://doi.org/10.7554/eLife.39636.026

• Transparent reporting form
DOI: https://doi.org/10.7554/eLife.39636.027

### Data availability

Sequencing data have been deposited to GEO under accession codes GSE111588.

The following dataset was generated:

| Author(s) | Year | Dataset title | Dataset URL | Database, license, and accessibility information |
|---|---|---|---|---|
| Gupta RK, Hepler C | 2018 | Single cell RNA-sequencing of visceral adipose tissue Pdgfr*β*+ cells | https://www.ncbi.nlm.nih.gov/geo/query/acc.cgi?acc=GSE111588 | Publicly available at the NCBI Gene Expression Omnibus (accession no. GSE111588) |

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
