## [Decision Letter]

[**Editorial note:** This article has been through an editorial process in which the authors decide how to respond to the issues raised during peer review. The Reviewing Editor's assessment is that all the issues have been addressed.]

Thank you for submitting your article "Identification of functionally distinct fibro-inflammatory and adipogenic stromal subpopulations in visceral adipose tissue of adult mice" for consideration by *eLife*. Your article has been reviewed by three peer reviewers, including Peter Tontonoz as the Reviewing Editor and Reviewer #1, and the evaluation has been overseen by Marianne Bronner as the Senior Editor. The other two reviewers remain anonymous.

The Reviewing Editor has highlighted the concerns that require revision and/or responses, and we have included the separate reviews below for your consideration. If you have any questions, please do not hesitate to contact us.

Summary:

Gupta and colleagues combine single cell transcriptomics with FACS to define the heterogeneity of a population of PDGFRβ-positive cells that reside in the adipose tissue stromal vascular fraction and are enriched for adipogenic precursors. They identify distinct clusters of cells that are postulated to represent adipocyte precursors, committed preadipocytes, fibro-inflammatory progenitors, and mesothelial-like cells. They further show that the fibro-inflammatory progenitor FIP pool lacks adipogenic capacity, and that this may be due to production of a secreted factor.

Essential revisions:

The individual reviews are included for the authors consideration. We understand that major additional in vivo work is beyond the scope of a revision. We request that the authors give particular attention to the following issues in preparing their revision.

1) It would be informative to include a comparison and discussion of the relationship between the cell populations described here and those identified in other recent scRNA-Seq papers on adipocyte precursors.

2) Please discuss the implications of the findings for visceral fat expansion as mentioned by reviewers 1 and 3.

3) Please consider whether it might be possible to use the new markers revealed by scSeq here to identify cell populations in tissue (e.g. by immunofluorescence) as suggested by reviewer #3.

Separate reviews (please respond to each point):

*Reviewer #1:*

Adipose tissue is remarkably heterogenous and various cell types in adipose contribute to its overall function. The authors had previously reported a population of PDGFRβ+ cells that give rise to white adipocytes in visceral fat. Using scRNA-Seq, the authors in this manuscript characterize various cell population within the PDGFRβ expressing cells. The manuscript reports FIP cell population that have highly compromised adipogenic potential and inhibits APCs adipogenic capacity in a paracrine-manner. The findings in the manuscript are interesting and the experiments are well-designed. Several questions that the authors might consider addressing are provided below.

Comments:

1) Although likely beyond the scope of the present study, it will be interesting to test the in vivo differentiation potential of APCs, FIPs, and MLCs by implantation.

2) Will the knockdown of NR4a members increase FIPs adipogenic potential?

3) Since NR4a is expressed by different cell types and had been previously shown to affect adipogenesis. Will deleting some of the identified genes in the FIP population (Figure 1 or Figure 2G) rescue the non-adipogenic phenotype?

4) Do the FIPs share some cellular and genetic identities with adipose CD142+/ABCG1+ SVF cells or vice-versa? These cells were also recently reported to be refractory to adipogenesis (PMID:29925944). Interestingly, CD142+cells were also shown to populate perivascular region in adipose.

5) In Figure 6A, FIP population increases and APC population markedly decreases under high-fat diet. Visceral fat expands by both hypertrophy and hyperplasia and APCs could be responsible for healthy adipose expansion and accounts for the majority of de novo adipogenesis within PDGFRβ+ cells. Please explain this paradox.

6) Will the conditioned media from FIPs also affect adipogenic potential of tdTomato expressing cells?

7) In Figure 3, the authors should also test the proliferative capacity of the isolated cells.

*Reviewer #2:*

The manuscript by Hepler et al. employed single-cell RNA sequencing and analyzed cellular heterogeneity in the stromal fraction of visceral adipose tissue. The bioinformatic analysis identified unique populations, including adipocyte precursor cells (APCs) and fibro-inflammatory progenitors (FIPs). The authors also established a robust sorting method to isolate APCs and FIPs by using their unique cell surface markers. Intriguingly, the authors found that FIPs acts on APCs to suppress their adipogenic potential via secretory factors, likely pro-inflammatory molecules. The authors further examined the extent to which a high-fat diet feeding affects the transcription profile of APCs and FIPs.

This is an outstanding paper that provides important insights into adipose progenitor heterogeneity and also useful information regarding the sorting method and the transcriptome data. Identification of Nr4a transcriptional factors suggests new regulators of FIPs. Overall, the data are convincing and the manuscript is well-written. Given the high significance and impact, this paper should be published in a timely manner. I have only minor comments that the authors wish to supplement in the paper.

1) A recent paper by Granneman's group (Burl et al., 2018) reported scRNA-seq analysis and defined subpopulations of Lin- epididymal WAT. It would be interesting to cross the dataset and if any common genes/pathways are seen.

2) TZDs are known to suppress inflammation and fibrosis pathways. Does TZD treatment inhibit pro-fibrosis and pro-inflammatory genes in FIPs?

3. The authors wish to comment on any difference in APC and FIP populations between male and female mice.

Additional data files and statistical comments:

I think the study is rigorous.

*Reviewer #3:*

The manuscript by Hepler et al. is part of a recent wave of studies using single cell RNA sequencing to define the adipocyte precursor cell population. Studies for many years prior suggest that adipocyte precursor pools are heterogeneous, but it has been difficult to define at the molecular level.

Here, the authors take advantage of recent advances in single cell transcriptomics combined with FACS to molecularly define the heterogeneity of a population of PDGFRβ-positive cells that they previously described, which reside in the adipose tissue stromal vascular fraction, and are highly enriched for adipogenic precursors. Base on gene expression signatures, they identify 4 distinct clusters of PDGFR-β expressing cells that in their model represent adipocyte precursors, committed preadipocytes, fibro-inflammatory progenitors, and mesothelial-like cells. They also find that the fibro-inflammatory progenitor pool (termed FIPs) lack adipogenic capacity, and have anti-adipogenic effects that may be linked to a secreted factor.

The study is timely, well done, interesting, and follows a logical course of investigation. The stated hypothesis is that PDGFRβ expressing cells in gonadal WAT are heterogeneous. This fairly safe hypothesis is supported by the data, and while these data are not overly surprising, they add important details to the molecular definition of what are currently considered to be "APC" pools. A particular strength of the study is the identification of unique markers that define each subpopulation of cells. The major weakness of the study is that it falls short of showing that these subpopulations, in particular the FIP population, are present and have anti-adipogenic potential in vivo. In summary, these data support the notion that current protocols used to isolate APCs result in highly heterogeneous populations, and will help in defining the hierarchy of adipocyte lineages, but await in vivo functional confirmation.

Major Points:

My major concern is that the clusters of cell types identified by sequencing and shown to have different functions in vitro have not been confirmed in vivo. The authors have nicely identified unique markers of the FIP population; could these markers be used to define the in vivo populations (without the caveats of isolating and purifying cells by FACS)? For example, can they distinctly localize the FIP, APC, and mesothelial-like cells? Could the authors show high in vivo expression of the NR4A receptors selectively in the FIP cells? Can the authors somehow confirm the role of NR4A in FIPs in vivo?

In a related point, the authors' previous work suggests that all of the cells they sequenced here should express, or have previously expressed, *Pparg* (e.g. it was stated that PDGFRβ+ cells in gonadal WAT can be identified by *Pparg* expression). It is thus curious that non-adipogenic FIPs would express PPARg. It seems there could be an interesting biological reason for this (e.g. identity switching dependent on nutrient availability or other stimulus), but also potential technical caveats triggered by the isolation and purification strategy itself. This is why I think it is important to show that PDGFRβ+ (*pparg*+) FIP cells exist in the SVF in vivo (e.g. by immunofluorescence or similar strategy) and perform some type of in vivo loss-of-function analysis to demonstrate that these cells are (1) non-adipocyte forming and (2) anti-adipogenic to the APCs.

Other points:

Could the authors comment on the PDGFRβ-negative population of stromal vascular fraction cells that were selected against? Are there any adipogenic cells in this population and could the bias towards the PDGFRβ pool have caused the authors to miss important adipocyte progenitors?

A recent study by Deplancke, Wolfrum and colleagues in Nature similarly identified a population of anti-adipogenic cells they termed Aregs. In contrast to FIPs, Aregs appear to represent a small fraction of the APC pool. Are FIPs similar to Aregs? The same? Could the authors compare their gene signatures and comment on this?

Regarding the current studies' rationale, it is stated (Introduction, second paragraph) that it has long been appreciated that individuals who preferentially accumulate WAT in subcutaneous regions are at lower risks for insulin resistance compared to those who accumulate visceral adiposity. It is stated in the third paragraph of the Introduction, that healthy WAT expansion occurs by hyperplasia. Yet, in the fourth paragraph of the Introduction, it is stated that (in mice) epididymal (visceral) WAT expands by hypertrophy and hyperplasia, but subcutaneous WAT almost exclusively by the stated unhealthy modality of hypertrophy. Thus, there is disconnect between subcutaneous being healthy but expanding by unhealthy means. Could the authors clarify this? It also raises the important question of whether subcutaneous WAT precursors also exhibit the same heterogeneity as the visceral WAT precursors described here by the authors. Could the authors show this and discuss their findings?

Using their gene expression data, could the authors show whether there is additional heterogeneity within the 4 clusters, or whether these populations represent fairly homogenous populations? This, is important towards the overall theme of "heterogeneity" in the APC pool and for helping readers understand how much the current strategy improves purity.

---

## [Author Response]

Gupta and colleagues combine single cell transcriptomics with FACS to define the heterogeneity of a population of PDGFRβ-positive cells that reside in the adipose tissue stromal vascular fraction and are enriched for adipogenic precursors. They identify distinct clusters of cells that are postulated to represent adipocyte precursors, committed preadipocytes, fibro-inflammatory progenitors, and mesothelial-like cells. They further show that the fibro-inflammatory progenitor FIP pool lacks adipogenic capacity, and that this may be due to production of a secreted factor.

We thank the reviewers and editors for their careful consideration of our manuscript and for the very constructive feedback. As a result of the review/revision process, we feel the manuscript has improved. Further below, we provide a point-by-point response to individual critiques; however, the major changes can be summarized as follows:

- Comparison of our results to the recently published results from other single-cell sequencing studies;

- Additional functional analyses of APCs and FIPs, including in vivo transplantation studies and in vitro cell proliferation assays;

- Analysis of perivascular cell heterogeneity in additional WAT depots; inclusion of female mice;

- Further discussion of how FIPs may impact visceral adipose tissue remodeling and unresolved questions.

Essential revisions:The individual reviews are included for the authors consideration. We understand that major additional in vivo work is beyond the scope of a revision. We request that the authors give particular attention to the following issues in preparing their revision.1) It would be informative to include a comparison and discussion of the relationship between the cell populations described here and those identified in other recent scRNA-Seq papers on adipocyte precursors.

We certainly agree that this is important. Deplanke/Wolfrum and colleagues defined an anti-adipogenic population of cells within inguinal WAT, termed Aregs (Schwalie et al., 2018). We now demonstrate that the visceral adipose FIPs described here are not enriched in the expression of markers used to define inguinal WAT Aregs. Despite some functional similarities (e.g. lack of adipogenic potential; anti-adipogenic), FIPs appear to be molecularly distinct from Aregs. These new data have been added to the new Figure 5.

Granneman and colleagues independently performed scRNA-seq of adipose SVF cells (Burl et al., 2018). The authors identified 2 prominent adipose stem cell (ASC) populations (termed ASC 1 and ASC 2) with the gonadal WAT depot. The identified populations were not isolated and explored functionally in their study; however, a comparison of the molecular profiles (now included here) strongly suggests that ASC 1 defined by the authors bears close resemblance to APC population defined in our study. Moreover, ASC 2 bears close resemblance to the FIPs discovered here. Importantly, the similarity in results between the two independent approaches suggest that the MuralChaser model captures a significantly large portion of the apparent visceral WAT adipocyte precursor pool, and that our new sorting strategy can be used to capture these cells. These new data are added to the Figure 3—figure supplement 1 of the revised manuscript.

2) Please discuss the implications of the findings for visceral fat expansion as mentioned by reviewers 1 and 3.

The reviewers ask a number of great questions here. We appreciate the opportunity to discuss this further in the revised Discussion section. Our responses to the individual reviewer critiques are provided below.

3) Please consider whether it might be possible to use the new markers revealed by scSeq here to identify cell populations in tissue (e.g. by immunofluorescence) as suggested by reviewer #3.

We thank the reviewers for bringing up this important question. The identification of APCs and FIPs through the use of the MuralChaser mice essentially places these cells in the perivascular region of adipose tissue. As shown in the manuscript and in our prior publications (Hepler et al., 2017a; Hepler et al., 2017b; Shao et al., 2018; Vishvanath et al., 2016), mGFP expression in these reporter mice is confined to perivascular cells and to a few cells within the mesothelial layer of visceral WAT. Nevertheless, a simple tool to identify the specific subpopulations is needed for the field and is essential for understanding the spatial relationship between FIPs and APCs in vivo.

We have tried hard over the past year to find antibodies that 1) recognize proteins enriched in either FIPs or APCs, and 2) are suitable for immunohistochemistry (paraffin embedded sections). Specifically, we tested multiple commercial antibodies from Abcam, Cell Signaling, and other vendors, against LY6C, CD9, and other potential markers (AGT, DACT2, NR4A1, etc.). We do in fact observe staining patterns suggestive of heterogeneity within the PDGFRβ+ cells; however, we simply do not have confidence in any of these antibodies as the concentrations needed to obtain signal are quite high and we lack genetic controls to confirm their specificity. We are still actively working on developing these tools. We now discuss this issue in the revised Discussion section.

Separate reviews (please respond to each point):

Reviewer #1:

Adipose tissue is remarkably heterogenous and various cell types in adipose contribute to its overall function. The authors had previously reported a population of PDGFRβ+ cells that give rise to white adipocytes in visceral fat. Using scRNA-Seq, the authors in this manuscript characterize various cell population within the PDGFRβ expressing cells. The manuscript reports FIP cell population that have highly compromised adipogenic potential and inhibits APCs adipogenic capacity in a paracrine-manner. The findings in the manuscript are interesting and the experiments are well-designed. Several questions that the authors might consider addressing are provided below.Comments:1) Although likely beyond the scope of the present study, it will be interesting to test the in vivo differentiation potential of APCs, FIPs, and MLCs by implantation.

This is great suggestion. We performed these studies by transplanting isolated APCs and FIPs into a recently described mouse model of complete lipodystrophy (Adiponectin-Cre; *Pparg^l^*^oxP/loxP^) (Wang et al., 2013). As expected in these types of assays, there was some degree of variability; however, in all four mice injected with APCs, adipocytes emerge and are readily detectable. On the other hand, transplanted FIPs did not give rise to adipocytes in any of the four mice tested. Even under these extreme lipodystrophic conditions, FIPs appear refractory to adipocyte differentiation. These new data have been added to Figure 3—figure supplement 3.

2) Will the knockdown of NR4a members increase FIPs adipogenic potential?

Good question, especially in light of literature defining a role for NR4a members in suppressing adipogenesis (Chao et al., 2008). Knockdown of NR4a1 indeed led to an increase in the expression of *Pparg* in FIPs; however, this appeared to be insufficient in rendering cells more adipogenic, at least under the differentiation conditions utilized. This may be due to the fact that other NR4a family members serve in this capacity (Chao et al., 2008). We have opted not to include this in the manuscript as we are developing animal models to address this more thoroughly. We do, however, discuss this issue in the revised Discussion section.

3) Since NR4a is expressed by different cell types and had been previously shown to affect adipogenesis. Will deleting some of the identified genes in the FIP population (Figure 1 or Figure 2G) rescue the non-adipogenic phenotype?

This is a great question. We have not yet performed a comprehensive functional screen of potential anti-adipogenic factors within FIPs (e.g. transcription factors) and/or secreted from FIPs (e.g. WNTs, TGFβ, etc.). This is the focus of ongoing/future studies in the lab.

4) Do the FIPs share some cellular and genetic identities with adipose CD142+/ABCG1+ SVF cells or vice-versa? These cells were also recently reported to be refractory to adipogenesis (PMID:29925944). Interestingly, CD142+cells were also shown to populate perivascular region in adipose.

This is an important question. We now demonstrate that the visceral adipose FIPs described here are not enriched in the expression of markers used to define inguinal WAT Aregs. Despite some functional similarities (e.g. lack of adipogenic potential; anti-adipogenic), FIPs appear to be molecularly distinct from Aregs. These new data have been added to the new Figure 5.

*5) In Figure 6A, FIP population increases and APC population markedly decreases under high-fat diet. Visceral fat expands by both hypertrophy and hyperplasia and APCs could be responsible for healthy adipose expansion and accounts for the majority of de novo adipogenesis within PDGFR*β*+ cells. Please explain this paradox.*

This is a great question. “New” adipocytes originating from *Pdgfrb*-expressing cells first emerge within visceral WAT depots after 4 weeks of HFD feeding. The fact that the frequency of APCs decline by this time point may suggest that they have gone down the differentiation pathway (and no longer actively express *Pdgfrb*) and by this point are possibly depleted. With these new tools in hand, we are now revisiting the pattern of visceral fat expansion in obese mice, with a closer focus on the temporal and spatial events occurring following on the onset of HFD-feeding.

Nevertheless, the reviewer is correct that the increase in the frequency in FIPs is also somewhat puzzling. On one hand, the increase in this population correlates with the pathologic features of WAT expansion occurring following HFD feeding (inflammation/fibrosis). On the other hand, one may expect that the increased frequency of FIPs would completely blunt the differentiation of APCs. One possibility is that the anti-adipogenic activity of FIPs (rather than the frequency per se) is shut-off by local signals. Another possibility is that the spatial relationship between the two cell populations in vivo ultimately impacts how the cells interact. As now described in the revised Discussion section, further functional studies of these cells in vivo, using existing and perhaps new genetic tools, will be needed to further clarify the importance of these distinct mural cell phenotypes.

6) Will the conditioned media from FIPs also affect adipogenic potential of tdTomato expressing cells?

We did not address this directly in our current study; however, we have observed that tdTomato expressing cells from gonadal WAT do not differentiate as readily as the mGFP+ cells in the model. This is not to say that other preadipocyte populations do not exist; the presence of immune cells, endothelial cells, and perhaps other tdTomato+ cell types in the SVF cultures may mask these cells, under the conditions we utilize.

7) In Figure 3, the authors should also test the proliferative capacity of the isolated cells.

We thank the reviewer for this suggestion. Consistent with our observations in vivo, FIPs appear to proliferate much more rapidly than APCs upon plating in vitro. These new data are added to the new Figure 3—figure Supplement 1.

Reviewer #2:

The manuscript by Hepler et al. employed single-cell RNA sequencing and analyzed cellular heterogeneity in the stromal fraction of visceral adipose tissue. The bioinformatic analysis identified unique populations, including adipocyte precursor cells (APCs) and fibro-inflammatory progenitors (FIPs). The authors also established a robust sorting method to isolate APCs and FIPs by using their unique cell surface markers. Intriguingly, the authors found that FIPs acts on APCs to suppress their adipogenic potential via secretory factors, likely pro-inflammatory molecules. The authors further examined the extent to which a high-fat diet feeding affects the transcription profile of APCs and FIPs.This is an outstanding paper that provides important insights into adipose progenitor heterogeneity and also useful information regarding the sorting method and the transcriptome data. Identification of Nr4a transcriptional factors suggests new regulators of FIPs. Overall, the data are convincing and the manuscript is well-written. Given the high significance and impact, this paper should be published in a timely manner. I have only minor comments that the authors wish to supplement in the paper.

We thank the reviewer for his/her positive feedback and suggestions.

1) A recent paper by Granneman's group (Burl et al., 2018) reported scRNA-seq analysis and defined subpopulations of Lin- epididymal WAT. It would be interesting to cross the dataset and if any common genes/pathways are seen.

This is a good suggestion. We discuss this issue above and repeat our response here for convenience:

Granneman and colleagues independently performed scRNA-seq of adipose SVF cells. The authors identified 2 prominent adipose stem cell (ASC) populations (termed ASC 1 and ASC 2) with the gonadal WAT depot. The identified populations were not isolated and explored functionally in their study; however, a comparison of the molecular profiles strongly suggests that ASC 1 defined by the authors bears close resemblance to APC population defined in our study. Moreover, ASC 2 bears close resemblance to the FIPs discovered here. Importantly, the similarity in results between the two independent approaches suggest that the MuralChaser model captures a significantly large portion of the apparent visceral WAT adipocyte precursor pool, and that our new sorting strategy can be used to capture these cells. These new data are added to the Figure 3—figure supplement 1 of the revised manuscript.

2) TZDs are known to suppress inflammation and fibrosis pathways. Does TZD treatment inhibit pro-fibrosis and pro-inflammatory genes in FIPs?

This is a great question. In fact, we recently published that the beneficial effects of TZDs on gonadal WAT remodeling depend on PPARγ expression in *Pdgfrb*-expressing cells (Shao et al., 2018). Specifically, inducible deletion of PPARγ selectively in PDGFRβ+ cells blocks the ability of TZDs to promote adipocyte hyperplasia and suppress WAT inflammation. Whether the latter is due to an impact on FIPs is a great question. We are pursuing this question as part of a separate and extensive study (i.e. new mouse models) that focuses more broadly on the role of FIPs in regulating WAT inflammation.

3) The authors wish to comment on any difference in APC and FIP populations between male and female mice.

This is an important point. The sorting strategy that we employ to isolate FIPs and APCs from visceral WAT can be used in both female and male mice. We added this new data to the new Figure 4—figure supplement 1.

Additional data files and statistical comments:I think the study is rigorous.

Reviewer #3:

[…] The study is timely, well done, interesting, and follows a logical course of investigation. The stated hypothesis is that PDGFRβ expressing cells in gonadal WAT are heterogeneous. This fairly safe hypothesis is supported by the data, and while these data are not overly surprising, they add important details to the molecular definition of what are currently considered to be "APC" pools. A particular strength of the study is the identification of unique markers that define each subpopulation of cells. The major weakness of the study is that it falls short of showing that these subpopulations, in particular the FIP population, are present and have anti-adipogenic potential in vivo. In summary, these data support the notion that current protocols used to isolate APCs result in highly heterogeneous populations, and will help in defining the hierarchy of adipocyte lineages, but await in vivo functional confirmation.Major Points:My major concern is that the clusters of cell types identified by sequencing and shown to have different functions in vitro have not been confirmed in vivo. The authors have nicely identified unique markers of the FIP population; could these markers be used to define the in vivo populations (without the caveats of isolating and purifying cells by FACS)? For example, can they distinctly localize the FIP, APC, and mesothelial-like cells? Could the authors show high in vivo expression of the NR4A receptors selectively in the FIP cells? Can the authors somehow confirm the role of NR4A in FIPs in vivo?

This is a great point, and certainly a question that arises from all single cell sequencing studies. As noted above, we have made several attempts to localize these cell populations using commercially available antibodies recognizing FIPs and/or APC-selective proteins. We observe heterogeneous expression patterns within PDGFRβ+ cells; however, we are not yet confident enough in the specificity of the antibodies to suggest their use by the field.

A number of lines of evidence (independent of cell fractionation/FACs) suggest that mural cell heterogeneity exists in vivo. In fact, what motivated this study is the initial observation made from *Zfp423*^GFP^ reporter mice (Gupta et al., 2012; Vishvanath et al., 2016). Through the use of immunohistochemistry, we found that GFP expression in the adipose tissue of these animals can be found in a *subset* of perivascular cells. This is in line with earlier work from Graff and colleagues indicating that a subset of mural cells express *Pparg* (Tang et al., 2008).

*In a related point, the authors' previous work suggests that all of the cells they sequenced here should express, or have previously expressed, Pparg (e.g. it was stated that PDGFRβ+ cells in gonadal WAT can be identified by Pparg expression). It is thus curious that non-adipogenic FIPs would express PPARg. It seems there could be an interesting biological reason for this (e.g. identity switching dependent on nutrient availability or other stimulus), but also potential technical caveats triggered by the isolation and purification strategy itself. This is why I think it is important to show that PDGFRβ+ (pparg+) FIP cells exist in the SVF in vivo (e.g. by immunofluorescence or similar strategy) and perform some type of in* vivo *loss-of-function analysis to demonstrate that these cells are (1) non-adipocyte forming and (2) anti-adipogenic to the APCs.*

This is an important point and we thank the reviewer for the opportunity to clarify. Our work, along with the work of others (primarily Graff and colleagues), demonstrate that PDGFRβ+ adipocyte precursors (i.e. the subpopulation of PDGFRβ+ cells with adipogenic potential) are enriched in *Pparg* and *Zfp423* expression. The reviewer is correct that FIPs express *Pparg* and *Zfp423*; however, the levels appear quantitatively lower in FIPs vs. APCs.

Regarding function, we have now performed transplantation assays of FIPs/APCs into lipodystrophic mice. As noted above, in all four mice injected with APCs, adipocytes emerge and are readily detectable. On the other hand, transplanted FIPs did not give rise to adipocytes in any of the four mice tested. Even under these extreme lipodystrophic conditions, FIPs appear refractory to adipocyte differentiation. These new data have been added to Figure 3—figure supplement 3.

We completely agree with the sentiment of the reviewer that the new populations described here, along with other recently identified populations from other groups, need to be explored further in vivo. We have a number of on-going animal studies that involve genetic manipulation of inflammatory and fibrogenic signaling cascades in PDGFRβ+ cells. We expect that these studies, when completed, will shed insight into the importance of these various mural cell phenotypes in adipose tissue remodeling.

Other points:Could the authors comment on the PDGFRβ-negative population of stromal vascular fraction cells that were selected against? Are there any adipogenic cells in this population and could the bias towards the PDGFRβ pool have caused the authors to miss important adipocyte progenitors?

This is a great question. We have not explored the PDGFRβ- pool in any great depth. As such, we certainly cannot rule out that other precursors populations exist. In fact, our approach will only identify precursor populations that express *Pdgfrb/rtTA* at the time of the pulse labeling. *Pdgfrb* expression declines as cell undergo differentiation into adipocytes. This means that cells even further committed to the adipocyte lineage (i.e. no longer express *Pdgfrb*) would not be captured through our analysis. Moreover, putative stem cell populations not yet expressing *Pdgfrb* may also be present and not captured. All that being said, it is notable that our approach captured most, if not all, the precursor populations recently identified in the study by Granneman and colleagues. Their group performed single cell sequencing of the SVF from gonadal WAT, allowing them to capture (in principle) both PDGFRβ+ and PDGFRβ- precursors. As we now highlight in the revised manuscript, our analysis appears to capture the adipocyte precursor populations defined in their study. This does not exclude any possibility that additional populations exist; however, the similarity in results between the two independent approaches suggest that the MuralChaser model captures a significantly large portion of the apparent visceral WAT adipocyte precursor pool, and that our new sorting strategy can be used to capture these cells. We now discuss this more thoroughly in the revised Discussion section.

A recent study by Deplancke, Wolfrum and colleagues in Nature similarly identified a population of anti-adipogenic cells they termed Aregs. In contrast to FIPs, Aregs appear to represent a small fraction of the APC pool. Are FIPs similar to Aregs? The same? Could the authors compare their gene signatures and comment on this?

As noted above, we certainly agree that this is important. Deplanke/Wolfrum and colleagues defined an anti-adipogenic population of cells within inguinal WAT, termed Aregs. We now demonstrate that the visceral adipose FIPs described here are not enriched in the expression of markers used to define inguinal WAT Aregs. Despite some functional similarities (e.g. lack of adipogenic potential; anti-adipogenic), Visceral WAT FIPs appear to be at least somewhat distinct from the inguinal WAT Aregs described. These new data have been added to Figure 5.

Regarding the current studies' rationale, it is stated (Introduction, second paragraph) that it has long been appreciated that individuals who preferentially accumulate WAT in subcutaneous regions are at lower risks for insulin resistance compared to those who accumulate visceral adiposity. It is stated in the third paragraph of the Introduction, that healthy WAT expansion occurs by hyperplasia. Yet, in the fourth paragraph of the Introduction, it is stated that (in mice) epididymal (visceral) WAT expands by hypertrophy and hyperplasia, but subcutaneous WAT almost exclusively by the stated unhealthy modality of hypertrophy. Thus, there is disconnect between subcutaneous being healthy but expanding by unhealthy means. Could the authors clarify this?

This is a good point, and we appreciate the opportunity to discuss this further here. Indeed, in male mice, the subcutaneous inguinal WAT depot expands almost exclusively by adipocyte hypertrophy. It should be noted that all of our studies, and almost all related studies, have been done with C57BL/6 mice. These animals are often utilized in our field because it is a model of pathologic diet-induced obesity. The inability of the subcutaneous WAT depot to expand in a healthy manner may contribute to the overall systemic phenotype of diet-induced obese C57BL/6 mice. In fact, engineered animal models exhibiting hyperplastic subcutaneous adipose expansion in obesity (examples: Adiponectin transgenic, mitoNeet transgenic, Glut4 transgenic) maintain insulin sensitivity despite becoming obese. What is certainly puzzling is that PDGFRβ+ cells isolated from this depot are highly adipogenic in vitro (as shown here), but are not activated in vivo to undergo adipogenesis, even when a *Pparg* transgene is expressed (Shao et al., 2018). As such, there appear to strong suppressive signals in the tissue microenvironment that restrain adipogenesis. Such signals may emanate from the recently identified Aregs, or from other cell types.

As the reviewer correctly points out, it is then somewhat surprising that visceral WAT depot expands by both cell hypertrophy and cellular hyperplasia. It is in our view that the level of de novo adipogenesis naturally occurring in this depot following the onset of HFD is beneficial, but insufficient to protect from pathological WAT expansion. This view is based on our recently published models in which allow for inducible expression or deletion of *Pparg* in PDGFRβ+ cells (Shao et al., 2018). Loss of *Pparg* in PDGFRβ+ cells leads to a loss of de novo adipogenesis from PDGFRβ+ cells in the visceral WAT depot of diet-induced obese mice; this exacerbates the pathologic remodeling of this depot (i.e. increased inflammation and fibrosis). However, driving further de novo adipogenesis from PDGFRβ+ cells through transgenic *Pparg* expression leads to a healthy expansion of visceral WAT (lower inflammation and small adipocytes).

This leads to the question raised by reviewer 1: Why don’t the FIPs block de novo adipogenesis from the mural cell lineage in this depot? As noted above, it is possible that the anti-adipogenic activity of these cells is somewhat shut-off in the setting of nutrient excess, or turned on during later HFD feeding timepoints. In order to address this, we first need to identify the putative anti-adipogenic factor. Importantly, and as noted in the manuscript, further in vivo studies of FIPs and APCs will be needed in order to clarify their exact roles in WAT remodeling in vivo. We now address this issue directly in the revised Discussion.

It also raises the important question of whether subcutaneous WAT precursors also exhibit the same heterogeneity as the visceral WAT precursors described here by the authors. Could the authors show this and discuss their findings?

Great point. Transcriptional programs of white adipocyte precursors are depot- dependent. The question raised by the reviewer here motivated us to ask whether similar functional heterogeneity exists amongst PDGFRβ+ cells within various WAT depots, and whether functionally distinct subpopulations could be selected for using the same FACS strategy described above. Indeed, the same three populations can be observed within the mesenteric and retroperitoneal depots of adult male mice, with LY6C- CD9- PDGFRβ+ cells representing the highly adipogenic subpopulation (new Figure 4A-H).

We also examined LY6C expression within PDGFRβ+ SVF cells obtained from the inguinal and anterior subcutaneous WAT depots. We previously demonstrated that the total pool of PDGFRβ+ cells from inguinal WAT is very highly adipogenic in vitro; however, remarkably, all PDGFRβ+ cells within the inguinal and anterior subcutaneous WAT depots expressed LY6C (new Figure 4I). These data suggest that if heterogeneity exists amongst PDGFRβ+ cells in these subcutaneous depots, they could not be discriminated on the basis of LY6C expression. Therefore, functionally distinct stromal populations from visceral, but not subcutaneous, WAT depots can be revealed on the basis of LY6C and CD9 expression.

On-going single cell sequencing studies in our lab are examining whether cell populations representing APCs and FIPs exist within subcutaneous WAT depots and brown adipose tissue depots, perhaps bearing distinct markers.

Using their gene expression data, could the authors show whether there is additional heterogeneity within the 4 clusters, or whether these populations represent fairly homogenous populations? This, is important towards the overall theme of "heterogeneity" in the APC pool and for helping readers understand how much the current strategy improves purity.

We agree that this is an important question. Through further Cell Ranger analysis of APCs and FIPs, we did not see any clear and obvious subdivision of the two populations. This does not mean that APCs/FIPs are not and/or cannot be heterogeneous under any circumstance (e.g. in obesity). We are interested in further evaluating heterogeneity of potential subpopulations of FIPs and APCs in future studies, through 1) single cell-derived clonal analyses in vitro, and 2) additional scRNA-seq analyses (re-sequencing) of purified FIPs and APCs. In the revised Discussion section we further emphasize the potential importance of further analyses going forward.